

# Implications of climate and litter quality for simulations of litterbag decomposition at high latitudes

Elin Ristorp Aas[1,2], Inge Althuizen[3,4], Hui Tang[1,5], Sonya Geange[4], Eva Lieungh[6], Vigdis Vandvik[4], and Terje Koren Berntsen[1,2]

[1]Department of Geosciences, University of Oslo, Norway
[2]Centre for Biogeochemistry in the Anthropocene, University of Oslo, Norway
[6]Natural History Museum, University of Oslo, Norway
[3]Department of Biological Sciences and Bjerknes Centre for Climate Research, University of Bergen, Bergen, Norway
[4]NORCE Norwegian Research Centre, Bergen, Norway
[5]Finnish Meteorological Institute (FMI), Climate System Research, Helsinki, Finland

**Correspondence:** Elin Ristorp Aas (ecaas@geo.uio.no)

**Abstract.** Litter decomposition is a vital part of the carbon cycle and is thoroughly studied both in the field and with models. Although temporally and spatially limited, litterbag decomposition experiments are often used to calibrate and evaluate soil models intended for use on large scales, coupled to a land model. We used the microbially explicit soil decomposition model MIMICS+ to replicate two high-latitude litterbag decomposition experiments of different spatial and temporal scales. We investigated how well the model represented observed mass loss in terms of the controlling factors climate and litter quality and their relative importance with time. In addition to default model forcing, we used measured and site-specific model-derived microclimatic variables (soil moisture and temperature), hypothesizing that this would improve model results. We found that MIMICS+ represented mass loss after three and six years well across a climatic gradient of Canadian sites, but had more variable results for one-year mass loss across a climate grid in Southern Norway. In terms of litter quality, the litter metabolic fraction was more influential on modeled mass loss than the carbon-to-nitrogen ratio of the litter. Using alternative microclimate sources led to up to 25 % more mass remaining and down to 20 % less mass remaining compared to the simulations using default model input. None of the input alternatives significantly improved results compared to using the default model setup. We discuss possible causes for our findings and suggest measures to better utilize short-term field experiments to inform microbially explicit decomposition models.

## 1 Introduction

During microbial decomposition of organic matter, carbon is distributed between soil reservoirs and the atmosphere, giving soil microbes a crucial role in the terrestrial carbon cycle. In a world with increasing atmospheric $CO_2$ concentrations, net primary production and soil conditions will change, ultimately determining if soils act as a source or a sink of atmospheric carbon. The large amount of carbon stored in high-latitude and alpine soils (Sørensen et al., 2018; Crowther et al., 2019) together with rapidly increasing temperatures (Johannessen et al., 2016) makes it extra important to understand these relationships in these



ecosystems (IPCC, 2023). Reducing uncertainties linked to the representation of decomposition processes in models is key to more confidently projecting future climate scenarios (Sulman et al., 2018).

Broadly speaking, decomposition rates are controlled by three main factors: substrate quality, microbial community properties, and physical environment (Aerts, 1997). The relative importance of these factors for decomposition rates has recently been debated (Bradford et al., 2016; Joly et al., 2023). While macroclimate may generally appear to be the most influential mechanism (Bradford et al., 2016), Joly et al. (2023) illustrated how macroclimate affects litter quality through plant composition, which further can influence decomposer communities and thus decomposition rates. It is therefore necessary to consider these factors in the context of each other, and not as independent mechanisms. Decomposition processes take place in stages based on the chemical composition of the remaining substrate at any time (Chapin et al., 2011). Throughout these different stages, the importance of the above-mentioned controls varies, and the duration of observational experiments is therefore important to consider.

Litterbag experiments are the most commonly used technique to determine decomposition rates in the field (Kurz-Besson et al., 2005; Halbritter et al., 2020). Bags containing native or standard litter are buried and collected at pre-determined time intervals, and from the mass loss one can estimate decomposition rates and investigate relationships related to climate and litter quality (Adair et al., 2008). Although some long-term ( $\geq 10$ years) experiments exist (Trofymow and the CIDET Working Group, 1998; Harmon et al., 2009), and methods using standard litter have been proposed to overcome the issue (Keuskamp et al., 2013), litterbag experiments often cover limited spatial and temporal scales, and can not be expected to cover all phases of decomposition. Findings from the Canadian Intersite Decomposition Experiment (CIDET) three and six years after burial suggest that large-scale climate (mean annual temperature and summer precipitation) together with the ratio of recalcitrant organic compounds (acid-unhydrolyzable residue (AUR)) to nitrogen, are dominating controls on decomposition rates for sites covering a broad climatic gradient in Canada (Moore et al., 1999; Trofymow et al., 2002). For first-year decomposition, the initial chemical composition of the added litter was a major controlling factor (Preston and Trofymow, 2000), indicating that the importance of climate increases, whereas the initial litter composition becomes less important with time. Generally, decomposition rates increase with temperature and precipitation in temperature-limited systems (Aerts, 2006). Yet, some studies have found negative correlations between precipitation and decomposition rates for example in very wet regions (Althuizen et al., 2018; Djukic et al., 2018; Sierra et al., 2017; Schuur, 2001). These contrasting findings from observational studies illustrate the dual role of soil moisture, where decomposition is limited in both wet (anaerobic conditions) and dry (or freezing) conditions, with an optimum somewhere in between. To better understand these mechanisms, soil decomposition models are valuable tools.

The new generation of soil carbon models includes explicit representation of microbial activity (Chandel et al., 2023). By incorporating functions of separate microbial groups it is possible to examine the influence of various factors on the decomposition rates, and ultimately on the carbon budget. In these models, decomposition rates are usually represented as functions of temperature, moisture, and substrate properties (Sierra et al., 2015; Chandel et al., 2023). The representation of microclimate (soil temperature and moisture) in soil models can come directly from observations or be derived from other models. Microclimate simulated by land surface models depends on the representation of above-ground vegetation, soil properties,



of atmospheric forcing. When using litterbag studies to validate and evaluate soil carbon models, the modeled microclimate should be representative of the actual environment in which the study takes place, especially for short-term experiments where local conditions may have a relatively larger influence compared to large-scale climatic patterns. Few studies have investigated variability related to the representation of soil microclimate, which is of interest as efforts are made to incorporate microbially

explicit soil models into land surface models for use in Earth System Models (ESMs, Chandel et al. (2023)).

In Aas et al. (2023) we presented the soil decomposition model MIMICS+ which provided reasonable carbon and nitrogen stock values compared to a database of Norwegian forest soil profiles. Here, we use MIMICS+ to investigate mechanisms at play during litterbag decomposition using data from two experiments; CIDET (Trofymow and the CIDET Working Group, 1998), and a short-term experiment performed in Norway (Vandvik et al., 2022; Telford et al., 2023). These experiments were

chosen as they were found to be suitable for testing decomposition processes at high-latitude ecosystems. We test the following hypotheses: (1) MIMICS+ adequately captures observed patterns in mass loss in terms of climate and litter quality; (2) MIMICS+ includes the processes found to be governing litter mass loss on short (12 months) and longer (6 years) timescales, and captures the shift/evolution of dominating controls through the stages of decomposition; (3) an improved microclimate from observations or site-specific land model configurations improves model predictions of litterbag mass loss.

## 2 Methods

### 2.1 The Canadian Intersite Decomposition Experiment

The CIDET study is a long-term litter decomposition experiment covering major ecoclimatic regions in Canada (Trofymow and the CIDET Working Group, 1998). 11 litterbags, each containing 10 g of one native litter type (10 litter types and 1 wood block) were buried at 21 sites during autumn 1992. Information about litter carbon:nitrogen (C:N) ratio and chemical

recalcitrance was known. Litterbags were then collected yearly, dried, and analyzed for mass loss. In this study we use reported mass loss across the 11 litter types after three (Moore et al., 1999) and six years (Trofymow et al., 2002) for nine of the sites (see Fig. A1b). We chose sites categorized as subarctic or boreal, and wet and mild sites in the cordilleran ecoclimatic region because of their large-scale climate resemblance to the VCG sites (Sect. 2.2). See Table A1 and Table A2 for details about the CIDET sites and litter types, respectively. Data from the CIDET study is used to examine overall model performance against a

well-established, relatively long-term experiment (hypothesis 1).

### 2.2 The Vestland Climate Grid

The Vestland Climate Grid (VCG) is a set of 12 semi-natural grasslands in southern Norway that are situated along natural temperature and precipitation gradients (Klanderud et al., 2015; Vandvik et al., 2020). The grid covers three temperature levels with growing-season (June–September) temperatures of 6.5°C (Alpine, ALP), 8.5°C (Sub-Alpine, SUB) and 10.5°C (Boreal,

BOR), and four annual precipitation levels (ca. 600, 1200, 2000, and 2700 mm), see TableA3 and Fig. A1c.



In this study, we use data from the control litterbags of a transplant study at the VCG (Telford et al., 2023; Vandvik et al., 2022). In the control treatment, native graminoid litter from each site was collected and analyzed for carbon and nitrogen content, then put into litterbags (10 x 10 cm) and buried at its place of origin in 2016. The bags were buried as soon as the snow had melted, and collected 1, 2, 3, and 12 months after burial. After collection, the bags were cleaned, dried, and analyzed

for mass loss. As the bags were added in early summer, not in fall, the experiment does not intend to replicate a realistic litterfall event, but rather give more general insights into decomposition rates at semi-natural grasslands and the effect of climatic conditions. Soil temperature and moisture was recorded with data loggers at approximately 5 cm depth throughout the experiment period (Vandvik et al., 2022; Telford et al., 2023). Unlike the CIDET sites, the sites in the VCG were not chosen to be ideal for litterbag experiments, but rather to study a range of ecological processes across gradients in temperature and

precipitation. Therefore, the VCG data provides a basis for a more experimental investigation into local conditions, and their impact on decomposition mechanisms (hypothesis 3).

### 2.3    Description of the soil decomposition model, MIMICS+

The litterbag experiments were replicated with the soil decomposition model MIMICS+ (Aas et al., 2023). MIMICS+ is a vertically resolved, microbially explicit model which represents the activity of saprotrophic bacteria and fungi (microbial

decomposers) by temperature and moisture dependent reverse Michaelis Menten Kinetics (Moorhead and Weintraub, 2018). Vertical soil layers follow the same structure as the Community Land Model (CLM), with increasing layer thicknesses with depth (Lawrence et al., 2019). Litterfall, mycorrhizal carbon allocation, and nitrogen deposition, in addition to soil temperature, moisture, and clay content is required to run MIMICS+. We used CLM in single-site mode to produce this information (Sect. 2.4). For the VCG sites we also use alternative representations of soil temperature and moisture for the years 2016–2017 when

the litterbag experiment was performed (Sect. 2.5). Here we provide an overview of processes and assumptions in MIMICS+ that are relevant for litter decomposition, a more detailed model description is found in Aas et al. (2023), and an illustration of the model is found in Appendix B.

In MIMICS+, decomposition rates are calculated based on biotic and abiotic conditions within each soil layer. The general form of the rMMK equation for soil layer j, where F is the decomposition rate ($\mathrm{gCm^{-3}h^{-1}}$) of saprotrophic group $SAP$

(bacteria or fungi) decomposing substrate $SUB$ (both prognostic variables with units $\mathrm{gCm^{-3}}$), is

$$F_{SUB,SAP,j} = \frac{V_{max}(T,\Theta)_j \cdot SAP_j \cdot SUB_j}{K_m(T)_j + SAP_j} \tag{1}$$

Where T and $\Theta$ represents soil temperature and moisture, respectively, and $V_{max}$ and $K_m$ are Michaelis-Menten parameters; maximum reaction rate and half-saturation constant, respectively. $V_{max}$ and $K_m$ are exponential functions of temperature following German et al. (2012), in the same manner as other MIMICS based models (e. g. Wieder et al., 2015; Kyker-Snowman

et al., 2020). Decomposition rates increase with temperature, but the effect is dampened by the fact that both $V_{max}$ (in the numerator) and $K_m$ (in the denominator) increases with temperature. Soil temperature also affects saprotrophic lifetimes by reducing turnover rates to a minimum when soil temperatures are below zero. The maximum reaction rate ($V_{max}$) is also affected by soil moisture by a bell shaped scaling function ($f(\Theta)$) with a value between 0.05–1 (Wieder et al., 2017; Sulman



et al., 2014). This modifier limits decomposition under both very wet (oxygen limitation) and very dry conditions, and has a
maximum value (no moisture limitation on decomposition) when the fraction of soil water saturation is 0.55. In conditions
where a high fraction of the soil water is present in ice form, the soil will be perceived as dry, because the frozen water is
unavailable for the microbes. See Appendix B for details about the decomposition equations and climatic modifiers.

The quality of the incoming litter affects model decomposition in two ways; through the C:N ratio and the chemical recal-
citrance. The C:N ratio determines how much of the carbon from the substrate the saprotrophs can use for growth, and how
much is respired, to match the C:N requirement of the microbial pools (assumed to be $CN_b$=5 for bacteria and $CN_f$=8 for
fungi). Saprotrophs decomposing substrates with high C:N ratios will build less biomass, in turn slowing down decomposition
rates (cf. Eq. 1), compared to saprotrophs in low C:N environments. The chemical recalcitrance of incoming litter determines
the distribution between the two model litter pools, which have different base decomposition rates (see Fig. B1). The metabolic
litter pool is assumed to be relatively labile and easily decomposable, while the structural litter pool is recalcitrant and rich in
lignin, and therefore slower to decompose. The metabolic fraction, $f_{met}$, is a function of the lignin:N ratio of the incoming
litter, and determines the partitioning between the two pools. When using litter forcing from CLM, the lignin:N ratio is derived
based on the Plant Functional Type (PFT) composition in the gridcell. For each PFT there is a prescribed lignin:N ratio in the
litter, hence the PFT composition determines the overall lignin:N ratio of the litter (see Appendix B3). The PFT distributions
used at sites in this study are given in Appendix D.

According to Angst et al. (2021), a substantial portion of soil organic matter comes directly from plants, omitting the
microbial pathway. Therefore, half of the incoming litter is directed to soil organic matter pools directly, some of which is
meant to represent leaching of easily soluble compounds. Hence, leaching is implicitly represented in the model, as a simplified
process affecting the incoming litter, not as an explicit sink for the litter pools.

## 2.4  Generation of forcing data with CLM

The input to MIMICS+ was provided by running a series of different simulations with CLM (tag: ctsm5.1.dev113-6-gd2cf309),
and reading monthly or daily records of variables from the CLM history files (Fig. 1). The main configuration used CLM with
fixed PFTs, but with active biogeochemistry (BGC) and prognostic vegetation state variables (leaf area index and canopy
height). These simulations represent a configuration typically used for global ESM simulations (Lawrence et al., 2019). To
further explore the sensitivities of MIMICS+ to the uncertainties of its input data, an alternative configuration of CLM with
prescribed leaf area index based on satellite remote sensing data (Satellite Phenology, SP), and no active biogeochemistry, is
also adopted.

For all sites (9 CIDET sites + 12 VCG sites) CLM was run in single-site mode using default surface data (PFT distribution,
soil texture, slope, and depth, see details in Table D1 and Table D2). Following standard CLM spin-up protocol (Lawrence et al.,
2019), the model was spun up by cycling 20 years (1901-1920) of atmospheric forcing based on the Global Soil Wetness Project
v3 (GSWPv3, https://hydro.iis.u-tokyo.ac.jp/GSWP3); first 300 years in "accelerated spinup" mode, then regular spinup until
the change between two consecutive time cycles were less than 1 %, which took between 900–2000 years. The historical period
was run with GSWPv3 forcing until the end of 2014 for the CIDET sites (with sampling period 1992–1998). For the VCG



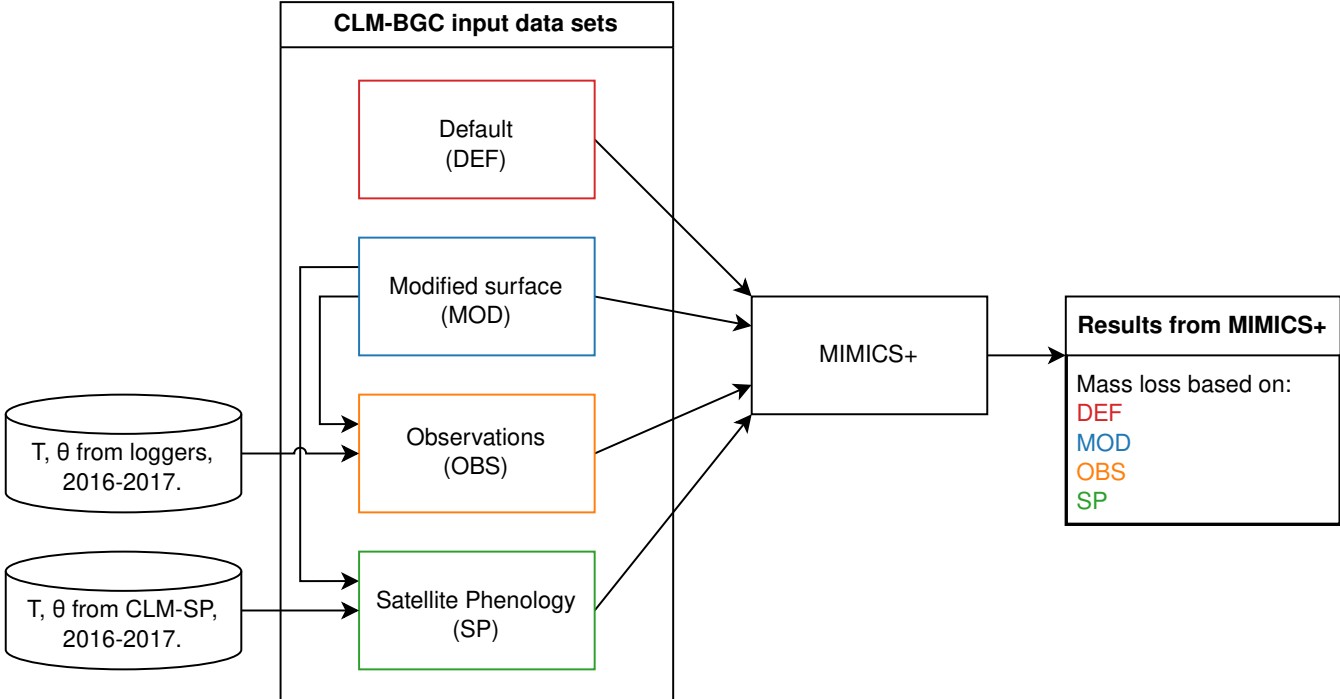

**Figure 1.** Illustration of the simulation setup using different input data for the VCG sites. All input data sets are based on CLM-BGC simulations, with either default surface parameters (DEF) of modified surface parameters (MOD). In the OBS and SP datasets, soil temperature (T) and soil moisture (Θ) for the years 2016–2017 are replaced with data from loggers and a CLM-SP simulation, respectively. Further details about the data sets are found in the main text (Sect. 2.4, Sect. 2.5).

sites (sampled during 2016–2017), the historical period from 1850–1994 was run with GSWPv3 forcing, while the remaining historical period (1995–2018) was forced with the regional reanalysis dataset COSMO-REA6 which has a resolution of 6 km
(Bollmeyer et al., 2015). Since the COSMO-REA6 data was only available up to and including 2018 we could not extend the simulation further. These simulations will be referred to as DEF (red box in Fig. 1).

For the 12 VCG sites, we also ran simulations with surface data modified to match observations of soil properties from the sites. These runs used 100 % natural vegetation of the PFT type "C3 grass", which was determined to best represent actual vegetation. As the measured soil depths were shallow, < 20 cm, the minimum soil depth allowed by CLM (40 cm) was used
in all simulations. Based on available observations we modified the organic fraction and surface slope parameters in CLM. Using locally observed slope is particularly important for accurate representation of soil water. Organic matter content affect soil thermal and hydrologic properties (for instance saturated water content). The CLM simulations using these modified, site-specific surface parameters will be referred to as MOD (blue box in Fig. 1, parameter values in Table D3).



## 2.5 Microclimate modifications

To test the effect of different sources of soil microclimate on decomposition we modified the variables soil temperature, soil liquid water and soil ice water in the MOD files for the years 2016-2017, when the VCG experiment took place. The saturated water content (porosity) was kept the same as in the MOD simulations, since we did not have suitable measurements. However, in CLM the saturated water content is determined by organic matter and sand content, so using the MOD simulations with observed organic matter content is partly taking local environment into consideration. The microclimate timeseries are
presented along with the results in Sect. 3.4.

To create alternative forcing based on observations, soil temperature and moisture in the soil layer where the litter was added, plus the layers directly above and below (three layers, covering soil depths 2-20 cm) were replaced by daily means of observed values from data loggers at approximately 5 cm depth. In the case of missing observations the monthly mean (based on measurements between 2008–2023) of the relevant month was used. Observed soil moisture was reported as volumetric soil water
content ($m^3 m^{-3}$), while the model moisture modifier requires a division between liquid water and ice water (see Appendix B). When the observed temperature was above freezing, we assumed all soil water to be in liquid form, while the fractions of liquid and frozen water from the original CLM simulations were used when observed temperature was below freezing. CLM uses an algorithm from Niu and Yang (2006) to determine the super-cooled liquid fraction in soils with temperatures below zero. Input data using observed microclimate will be referred to as OBS (orange box in Fig. 1).

The CLM SP simulations were created in connection to a model study of mosses at the VCG sites (Tang et al. in prep.). The soil depth, organic matter, and slope values for these simulations were modified in the same manner as described above, while the PFT was 100 % C3 arctic grass. The use of satellite phenology leads to different onset of spring and autumn conditions, and the prescribed leaf area index and canopy height affects insulation and thereby soil microclimate, making them different from the BGC simulations. For creating alternative forcing based on the SP simulations, daily soil temperature and moisture
from these history files were used (for the entire soil column). Note that since we were interested in the isolated effect of soil microclimate, we did not change litter input, drainage etc. for the alternative forcing. These input data will be referred to as SP (green box in Fig. 1).

The MIMICS+ results based on the four different input data sets (DEF, MOD, OBS and SP) are used to investigate how microclimate representation affects modeled predictions of litterbag mass loss (hypothesis 3).

## 2.6 Litterbag experiments in MIMICS+


MIMICS+ was spun up to equilibrium from arbitrary initial conditions by recycling input from the CLM simulations (DEF or MOD) for the years 1850–1869 for 1000 years. Then, the historical period was run using forcing data from the CLM simulations described above. The MIMICS+ simulations using MOD, OBS, or SP input data were continued from the spinup based on MOD, while the MIMICS+ simulations using DEF were continued from the DEF spinup. This gave initial decomposer
biomass values at the sites between 8–47 $gCm^{-2}$, accounting for 0.3–2.3 % of the total soil carbon in the layer where litterbags were added.



The modeled litterbags consisted of 10 gC and were added as a pulse to the third layer (covering depth 6-12 cm) of a soil column with an assumed area of 1 m² corresponding to an initial perturbation in the litter layer carbon concentration of 170 gCm⁻³. We assumed that the litterbag carbon stayed in the same soil layer throughout the experiment (except what is lost through respiration), and calculated the carbon mass loss by subtracting litter values in a control simulation (without added litter) from litter values in the experiment simulations. Since the litterbags were cleaned and dried after collection and before weighing, we assume minor contributions from soil organic matter and microbial biomass to the collected litterbags, and therefore only consider mass in the litter pools.

For the CIDET sites, simulations were run for all 11 litter types which differed in C:N ratio, and metabolic fraction (see Table A2). The metabolic fraction ($f_{met}$) was calculated using published measurements of lignin:N ratio with the same method as used for regular input litter, described in Appendix B. The litterbags were added on 10 October 1992 at all CIDET sites. As the litter in the VCG experiment came from graminoid species only, it was assumed to be relatively labile. The modeled litterbags were therefore added to the metabolic litter pool only. The N content was determined by measured C:N ratios of the collected litter at each site (Table A3). The litterbags for the VCG sites were added on the dates reported in Telford et al. (2023), between 18 May 2016 and 30 June 2016 for the twelve sites.

## 3 Results

### 3.1 CIDET site simulations

Generally, MIMICS+ replicated mass loss of the different litter types well for the CIDET sites (Fig. 2), and were within the observed standard deviations across all litter types for all but one site (NEL; Fig. 2c). Compared to the standard deviation of the observations, the spread between model simulations is small. The spread between simulations at sites with different macro-climates is larger than the spread between simulations of different litter types, indicating that climatic conditions dominate mass loss rates compared to litter quality in the model simulations. Low-quality litter, both in terms of C:N ratio and chemical recalcitrance, generally decompose slower than high-quality litter in the model, and the ranking among the litter types does not change throughout the simulations. The spread between simulations of different litter types tends to increase with temperature, indicating that litter quality is more important for litter mass loss in warmer regions. Modeled mass loss at the sites with the highest mean annual temperatures (PMC, SHL) have a sharp decrease from the beginning, and does not slow down significantly during winter, leading to the largest mass losses among the sites.

### 3.2 Relative litter mass loss at the CIDET sites

The higher metabolic fraction of jack pine (*Pinus banksiana*, coniferous evergreen), plains rough fescue (*Festuca halii*, grass), and trembling aspen (*Populus tremuloides*, broadleaf deciduous) litter leads to higher relative mass loss throughout the simulation compared to Douglas fir (*Pseudotsuga menziesii*, coniferous evergreen) and American beech (*Fagus grandifolia*, broadleaf deciduous) litter (Fig. 3). Initially, fescue is the fastest decomposing litter but with time jack pine, with the lowest C:N ratio,





**Figure 2.** Percent remaining mass in litterbags at CIDET sites. The lines show model simulations, with the color indicating the type of litter. The lighter the line color, the lower the C:N ratio (see Table A2). The dots show observed mean mass loss of each site across all litter types (Moore et al., 1999; Trofymow et al., 2002) with the error bar showing the standard deviation (SD). The SD was calculated from the reported standard error of the mean with N = 11. The inserted bars show the 1951–1980 average MAP (blue) and Jun–September average temperatures (orange) for each site (Trofymow and the CIDET Working Group, 1998).



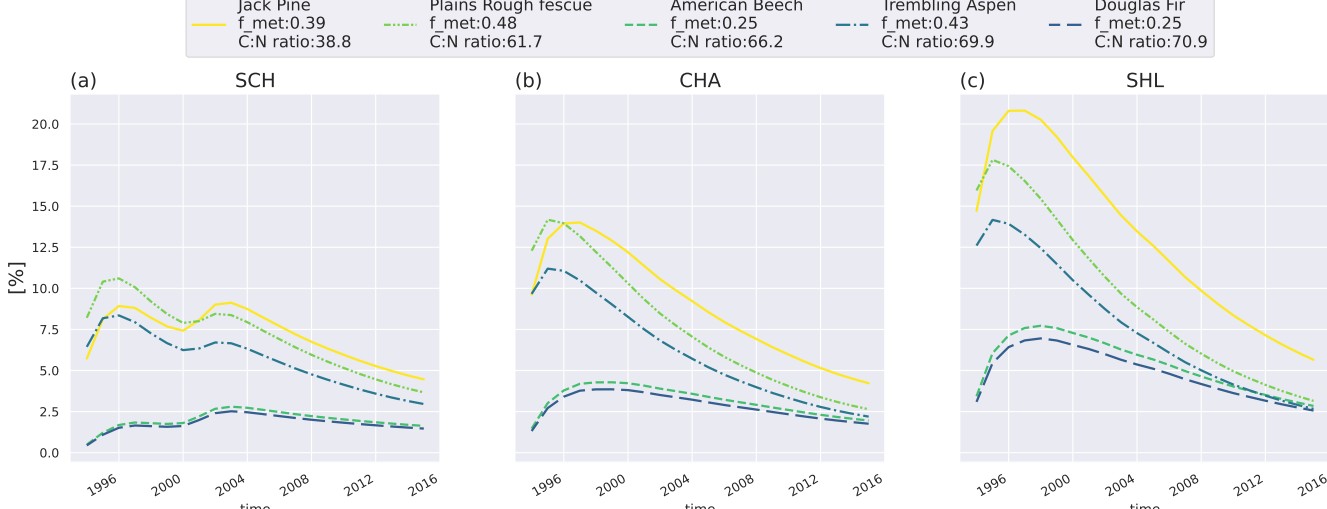

**Figure 3.** Simulated difference in cumulative mass loss since start of litterbag experiment of five litter types compared to mass loss of the slowest decomposing substrate, western hemlock (for western hemlock, f_met = 0.25 and C:N = 197.2). The values are normalized (%) to the cumulative mass loss of western hemlock. Results shown for the sites Schefferville (a, SCH), Chaplau (b, CHA) and Shawnigan lake (c, SHL). Details about sites and litter types are found in Appendix A. All sites and litter types are plotted in Fig. E1.

becomes the fastest decomposing litter. The time it takes before the relative mass loss of jack pine becomes greater than the mass loss of fescue is longer for the colder SCH site (ca. 10 years) than for the warmer CHA site (ca. 4 years) and the warmest
SHL site (ca 1 year). Douglas fir and American beech have C:N ratios comparable to trembling aspen, but decompose slower due to the low metabolic fraction. The difference in C:N ratio between American beech and Douglas fir leads to slightly higher relative mass loss for American beech than for Douglas fir.

### 3.3 VCG simulations

The model simulations agree well with observations for the warmer and wetter sites, especially SUB3, SUB4 and BOR3
(Figures 4g, 4h, 4k), and moderately well for the cold and drier sites, ALP1-3 (Fig. 4a–c), but underestimates mass loss at the warm and dry sites (SUB1–2, BOR1–2, Fig. 4e–f,4i–j) and the coldest and wettest site (ALP4, Fig. 4d). Modeled mass loss rates were relatively constant during the summer months, then decreased to nearly no loss during the winter months. For the sites where the model performs best, there was a similar pattern in the observations between the 3-month collection point and the 1-year collection point, while for the sites with the highest disagreement between observations and model (SUB1, SUB2,
BOR1, BOR2, Fig. 4e–f, 4i–j) the observed mass loss between these two collection points was substantial.



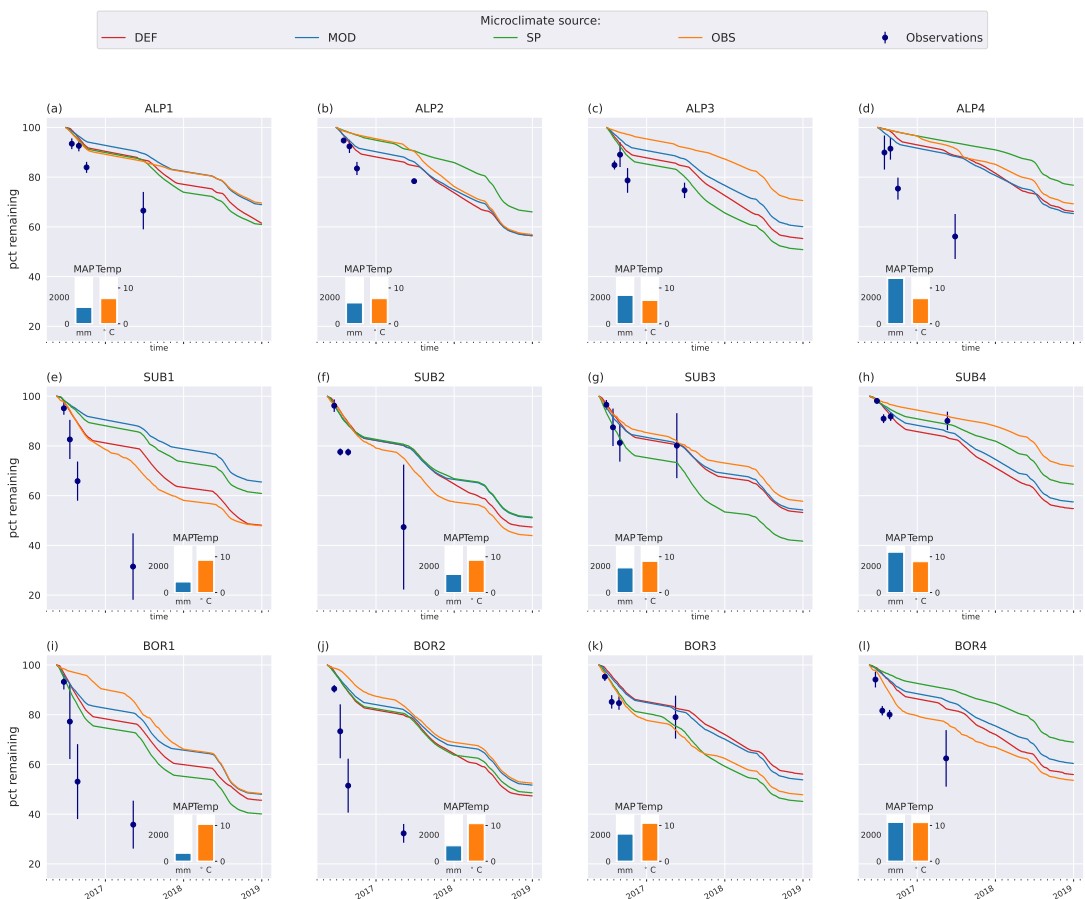

**Figure 4.** Remaining mass (%) in litterbags at VCG sites (Vandvik et al., 2022) for MIMICS+ simulations (solid lines) and observations (dots). The inserted bars show MAP (blue) and Jun–September average temperatures (orange) for each site (1960-1990 climate normal).



## 3.4 Effect of microclimatic forcing at VCG sites

Figure 5 shows soil temperature in the model layer where the litterbag was added, together with observed and modeled mass loss for the VCG sites. The general seasonal pattern is similar for all four microclimate sources, but the onset of spring and fall conditions differs markedly for some sites (e.g. SUB1, ALP4). The soil logger temperatures (OBS) are generally higher in fall
and winter than the model-generated soil temperatures, often remaining above freezing for a large part of the winter season.

Soil moisture varied more between the microclimate data than temperature (Fig. 6). Generally, the observed soil moisture is lower than modeled, especially for the sites with less precipitation (denoted by 1 and 2, e.g. ALP1/2 compared to ALP3/4). Model-based summer moisture is generally similar, while there is a larger variability during winter.

Using alternative microclimate sources led to up to 25 % more mass remaining (MOD for SUB1) and down to 20 % less
mass remaining (SP for SUB3) compared to the simulations using DEF input at the end of 2017 (when the alternative input data ends). Generally, simulation of the driest and wettest, warmer sites (SUB1, BOR1, SUB4, BOR4) showed the largest spread among the simulated mass losses (Fig. 4). There was no clear trend among the predictions; for example, simulations using observed microclimate returned the lowest mass loss rates for five sites (covering all macroclimate levels in the climate grid), but among the highest mass loss rates for three sites (SUB1, SUB2, BOR4). The predictions based on microclimate from
DEF and MOD are generally similar, except for SUB1, where the DEF simulation shows markedly higher mass loss rates. For the sites with the highest disagreement between model and observations (SUB1, SUB2, BOR1, BOR2), even the simulations with the highest loss rate significantly underestimated mass loss.

## 4 Discussion

In this study we used the soil model MIMICS+ to examine the representation of litter mass loss by replicating two litterbag
experiments; the relatively long-term CIDET study and the short-term VCG study. Results from the CIDET simulations showed that the model structure and parameters of MIMICS+ are able to reproduce observed decomposition mass loss from litterbag experiments at high latitudes. We found higher mass loss at warmer sites than at colder sites, in line with observations. Modeled mass loss was more sensitive to site conditions than litter type, but the spread between different litter types increased with increasing temperature (Fig. 2). The metabolic fraction was the most important litter quality control initially, while the C:N
ratio became more influential over time (Fig .3). Using different microclimate input data in the VCG simulations gave from 20 % less to 25 % more mass remaining at the end of 2017 (about 1.5 years after the litterbags were added) compared to the default (DEF) simulation. We found no clear trend or improvement among the simulations using different microclimates, however the winter slowdown in decomposition was less pronounced when using soil microclimate based on observations (Fig. 4).

### 4.1 Climate and litter quality controls on decomposition

MIMICS+ generally predicted higher mass loss at warmer compared to colder CIDET sites. Macroclimate (difference between sites) had a larger effect on mass loss than litter quality, with faster decomposition at warmer and wetter sites. This is in line







**Figure 5.** Left axis: Temperature in soil layer where litter is added from the four microclimate sources MOD, DEF, SP and OBS (see Fig. 1). Right axis: remaining litter from observations (black triangles, Vandvik et al. (2022)), simulations using DEF microclimate (red pentagons), MOD microclimate (blue squares), SP microclimate (green circles) and OBS microclimate (orange diamonds). The solid vertical line mark the burial date of the litterbag while the dotted vertical lines mark collection dates after 1,2,3 and 12 months.



**Figure 6.** Left axis: Soil moisture in the layer where litter is added from the four microclimate sources MOD, DEF, SP and OBS (see Fig. 1). Right axis: remaining litter from observations (black triangles, Vandvik et al. (2022)), simulations using DEF microclimate (red pentagons), MOD microclimate (blue squares), SP microclimate (green circles) and OBS microclimate (orange diamonds). The solid vertical line mark the burial date of the litterbag while the dotted vertical lines mark collection dates after 1,2,3 and 12 months.




with Trofymow et al. (2002) who found that MAT and (summer) precipitation were the controlling factors for mass loss after three and six years. The relative difference in mass loss between litter types was smaller at the colder sites, compared to the warmer sites (Fig. 3).

In MIMICS+, litter quality in terms of metabolic fraction has an immediate effect on decomposition rates, as the calculated metabolic fraction determines how much of the litter is treated as labile (LITm) and how much is treated as recalcitrant (LITs), which in turn affect the relative roles of saprotrophic bacteria and fungi. The C:N ratio affects modeled decomposition rates gradually, by affecting saprotrophic CUE, and thereby their growth rates. This separation is seen in Fig. 3, where the relative accumulated loss of the litter type jack pine, with a lower C:N ratio, eventually surpasses plains rough fescue with the highest metabolic fraction. Generally, the metabolic fraction had a larger influence on modeled litter mass loss rates than the C:N ratio (Fig. 3 and Fig. E1).

The six litter types with the highest mass loss, and the five types with the lowest mass loss, were the same for the MIMICS+ simulations and the CIDET observations after three and six years. However, the internal ranking within these two groups was different. The litter type that differs the most from the observed ranking is jack pine, which had the highest modeled mass loss after three years but only ranks as the 5th highest mass loss based on observations (Preston and Trofymow, 2000). As jack pine is a common species across Canada (Moore et al., 1999), the slower observed decomposition is likely not due to the introduction of an unknown substrate to the decomposer communities, which have been shown to affect decomposition (Joly et al., 2023). This coniferous, evergreen litter type is rich in lignin, but due to the high nitrogen content, the metabolic fraction used in the model is relatively high (see Table A2 and Eq. B4). This, together with the low C:N ratio leads to fast decomposition of jack pine throughout the simulations. The discrepancy with the observations indicates that the litter nitrogen content might have a too large influence on the estimated metabolic fraction in the model parameterization, where the metabolic fraction is a function of the lignin:N ratio.

As MIMICS+ reproduced mass loss variation with large-scale climate patterns for the CIDET sites within the observed variation for 8 of the 9 sites, and broadly captured the mass loss ranking among litter types, we conclude that MIMICS+ performs in line with hypothesis 1. This shows that the model framework is generally suited to capture mechanisms governing mass loss during the intermediate stage of litter decomposition (3–6 years).

## 4.2 Governing processes on different time scales

Trofymow et al. (2002) found that the chemical recalcitrance of litter (AUR and AUR:N ratio, comparable to lignin and lignin:N ratio) was more important for first-year mass loss than climatic variables. Although we see an immediate effect of chemical recalcitrance in the modeled results for the CIDET sites (Fig. 3), the absolute modeled mass loss was still controlled by climatic factors during the initial year. Mass loss at the warmest site was more than twice as high as the coldest site during the initial year, while the initial mass loss of the fastest decomposing litter type compared to the slowest was about 16 % higher (Fig. 3c). Sites that experienced freezing soil conditions during the first winter had a notably lower mass loss compared to sites without freezing winter conditions (Fig. 2). The effect of freezing is further discussed in Sect. 4.3.



The high observed mass loss at SUB1-2 and BOR1-2 at the VCG sites is likely related to favorable climatic conditions combined with low C:N ratio litter (high quality). Unpublished results from the VCG sites (Althuizen et al.) show a significant effect of litter C:N ratio on decomposition rates when litter was buried at warmer and/or wetter sites. As discussed for the CIDET sites, the effect of litter C:N ratio was small compared to the effect of the metabolic fraction of the litter, especially during the first year (which is the timescale of the VCG experiment). Together, these results call for further studies of the

effects of litter C:N ratio in combination with microbial stoichiometry. In harsh ecosystems like the locations of the VCG sites, microbes might have adjusted to work efficiently with a lower C:N ratio (Mooshammer et al., 2014). Although the modeled percentage of total carbon in microbial biomass where within the typically reported value of 1–3 % (Chapin et al., 2011), the modeled resource imbalance between substrate and microbial stoichiometric demand might be inaccurate, leading to discrepancies between modeled and realistic decomposition rates. In addition, the VCG sites are subject to grazing by

livestock, which might lead to increased nutrient availability in the soils, giving more favorable conditions for the microbial communities. As grazing processes are not included in CLM or MIMICS+, the nutrient availability, and thus the amount of saprotrophic decomposers might be underestimated.

       Although we assumed that all litter at the VCG sites was labile ($f_{met} = 1$), MIMICS+ still underestimated mass loss at several VCG sites. In addition to the effect of favorable conditions mentioned above, leaching processes likely also contributed

to the observed mass loss, and might explain some of the discrepancies between modeled and observed values. MIMICS+ simplified representation of leaching only affects the incoming litter fluxes, so the experiments as they are set up in this study will not capture this mechanism, as the litterbags were added directly into the model litter pools. Bokhorst et al. (2010) studied decomposition in Swedish sub-arctic heath-land soils, which are comparable to the VCG sites. Their experiment suggested that the majority of the initial mass loss of fresh autumn litter happened during the first few weeks, mainly due to leaching of

organic compounds, not microbial decomposition. For model information and evaluation purposes, a measure of how much of the mass loss can be attributed to leaching, and how much is actual microbial decomposition would be valuable. MIMICS+ (and other models) would benefit from representing leaching in a more refined way, also taking lateral flow of Dissolved Organic Carbon (DOC) into account. This would be valuable when coupling to a larger land model, as tracking of DOC is important for several reasons, e.g. correctly simulating responses to climate change and impacts on aquatic ecosystems.

Overall, MIMICS+ performs better at longer timescales (> 1 year), where climatic controls are assumed to dominate the decomposition rates. This is seen by the representation of mass loss at the CIDET sites, with a clear trend of higher mass loss at warmer sites. Although we see an initial effect of litter quality at the CIDET sites, climatic conditions were still the controlling factor. An observed effect of C:N ratio at the VCG sites was not found in the model predictions. Hypothesis 2 is therefore partly proven, as we could identify the control of climate on longer scales, but the effect of litter quality on short scales was smaller

than expected. A better representation of initial leaching, together with locally observed microbial stoichiometry would likely improve model performance on shorter timescales. Additionally, as the decomposition process is sensitive to local conditions, good quality measurements of microclimate are particularly important.



### 4.3 Implications of model microclimate representation

The way the modified microclimate simulations were performed in this study is quite simple. For the OBS and SP simulations,
we only modified the variables during the decomposition experiment (after the litterbags were buried), so any long-term effects of the different microclimates in terms of potential differences in e.g. initial microbial biomass are thus not captured in these experiments. The DEF and MOD simulations were spun up from different conditions, and therefore have different histories. Yet, the setup allowed us to isolate the immediate effect of soil climate on the decomposition process, and examine the effect of the real-time soil temperature and moisture during the experiment.

Local microtopography affects microclimate and thus decomposition rates. We therefore expected that input data based on locally measured values and/or improved surface characterization would improve model predictions at the VCG sites. To estimate the performance of the four simulation sets, we calculated the normalized root mean square error based on all observations (four sampling times) at all 12 sites (N=48, see Appendix E2) to be 0.27 (DEF), 0.28 (SP) and 0.29 (MOD and OBS). In other words, the model predictions from all simulation sets deviate from the observed values by 27-29 % of the total
range of the data, and we are not able to conclude that customized input data improved model predictions of mass loss in line with hypothesis 3. Although not having the expected effect on model results, the use of different microclimate sources still demonstrates several temporal and spatial challenges that arise when connecting locally observed quantities to a model framework.

At some VCG sites, for one or more of the four simulation sets, the modeled mass loss rate during high activity summer
season was similar to the observed mean annual loss rate (e.g. ALP1, SUB2, BOR1), indicating that modeled climatic inhibition of decomposition during the cold season is too strict, contributing to the underestimation of annual loss rate. Using observed soil temperature to determine ice fraction (as described in Sect. 2.5) resulted in markedly lower ice content than predicted by the CLM simulations (Fig. C3), also hinting that the real sites might not have been as limited by freezing water and/or low temperatures as predicted from CLM. For example, the high mass loss observed at the cold and wet site ALP4 may be due to
snow insulation keeping the soil unfrozen (Rixen et al., 2022). It has been shown that accurately representing snow in CLM is challenging, which can contribute to inaccurate representation of soil microclimate during winter (Aas et al., 2017). Contrary to the above-mentioned sites, observations at SUB3 and BOR3 show a winter slow-down of decomposition more in line with the modeled predictions. This distinction between observed winter decomposition shows that local conditions may allow for, or limit decomposition during winter, but even using locally measured temperature and moisture (OBS) did not improve the
modeled result. Snow melt-out dates in alpine ecosystems can also vary significantly even at small spatial scales (Pirk et al., 2023). This is likely affecting conditions at the VCG sites, where local features such as rock formations can be influential on snow patterns.

To examine how much the moisture limitation affected modeled mass loss, we ran a set of simulations where $f(\theta) = 1$ at all times, that is no moisture limitation on decomposition rates (Fig. C4). For the warmer and drier VCG sites (SUB1,
SUB2, BOR1, BOR2) this brought model predictions closer to observations, but still substantially underestimated mass loss. For the sites where the model initially performed well (SUB3, SUB4, BOR3) the simulations with no moisture limitation





overestimated mass loss after one year, indicating that local conditions not captured by model equations (and their dependency on temperature and moisture) play a dominant role at the sites on these time scales. The spread between simulations using different microclimates was smaller in experiments with no moisture limitation (Fig. C4), illustrating that the direct temperature

effect on decomposition was generally similar in the different representations (Fig. 5) while the moisture representation was more variable (Fig. 6). However, the interacting effect between temperature and moisture around the freezing point has a large impact on decomposition rates, and differences in ice fraction caused large differences in decomposition limitation due to freezing. Observed soil temperature at the SUB2 site was above zero for long periods during winter (Fig. 5f), while modeled winter soil temperatures were mainly below zero. This resulted in less moisture limitation for the MIMICS+ simulations using

observed microclimate, which led to a significantly higher mass loss after one year (Fig. C1f). This example illustrates how decomposition in ecosystems that are experiencing cold seasons is extra sensitive, not only to the isolated effects of temperature and moisture but also to the phase change of water (Sierra et al., 2015).

The generally higher soil moisture values from the model-derived microclimate compared to the observational data resulted in moisture limitations for different reasons. Decomposition in the simulations using observation-based microclimate was

mostly limited due to dry conditions, while the simulations using model-based microclimate were often limited due to high soil moisture (Fig. C2). This demonstrates the chance of getting model results right for the wrong reasons based on inaccurate input data. The use of data loggers placed at the observational sites reduces this chance, but also introduces their own uncertainty (Graae et al., 2012).

The VCG sites are open grasslands, while the CIDET sites are mostly forested. Microclimate at the CIDET sites is therefore

likely modified by the temperature buffering effect of forest canopies (De Frenne et al., 2019) more than the VCG sites. The VCG sites are also dominated by fast-growing species with relatively high C:N ratio (see Table A3). Together these factors could explain some of the differences in mass loss between the two observational datasets, where litterbags at some VCG sites loose more mass in one year than litterbags at the CIDET sites loose in three years.

### 4.4 Limitations and outlook

The individual observed litterbags in the experiments are subject to local conditions not captured by the modeled grid-cell averages. This is illustrated by the simulations of the CIDET sites and litter types, where modeled variability between simulations of different litter types was generally smaller than the variability indicated by the standard deviations of the observations. The main intention for MIMICS+ is to be incorporated and used within a land model framework on large/long regional and temporal scales, for which general parameters representative of larger areas are desirable. However, for investigating processes

and mechanisms on a local scale, efforts to gather site data suitable for informing model parameters and parameterizations, like microbial biomass and its chemical composition, should be prioritized.

Another local factor not considered in the model is the contribution of micro, meso, and macrofauna to the decomposition process. Litterbag mesh size has a significant effect on decomposition rates (Peng et al., 2022). The mesh size of the bags used in the VCG study (0.28 mm) allowed for microfauna, as well as some mesofauna to enter the bags in addition to microorganisms.



This might contribute to the model's underestimation of mass loss, especially at the sites where biodiversity is expected to be higher due to favorable climatic conditions (BOR1-2, SUB1-2).

The effect of using local slope and organic content data to produce input data for MIMICS+ (MOD) had a small effect on modeled mass loss (Fig. 4) compared to using default values (DEF). Since the model soil columns are deeper (at least 40 cm) than the observed soils ($< 20$ cm), hydrological processes could evolve quite differently in the model compared to the

observations and are probably contributing to the discrepancy between modeled and observed mass loss.

The sensitivity of model results to terms in the moisture function highlights the challenges of parameterizing soil moisture effects on decomposition rates. Sierra et al. (2015) showed that biogeochemical models use a range of different functions to represent moisture limitation. They emphasized the importance of representing limitation both due to oxygen limitation and dry conditions, as well as the effect of water phase change on microbial activity and decomposition rates. The moisture

function used in this study (Sulman et al., 2014; Wieder et al., 2017) is formulated in a way that can capture these effects; it includes one term representing liquid water deficit and one term representing air-filled porosity, in addition to separating between water in ice and liquid form. However, as our results show, it is highly dependent on good quality data on soil texture and ice formation and melting, and has a large influence on soil moisture and the decomposition rates in MIMICS+. Since snow cover has an important isolating effect and thereby affects soil freezing processes, this is a further motivation to improve

snow depth representation in CLM and other land surface models. Our results highlight the importance of accurate site-level information on soil properties to the performance of MIMICS+ (and other soil models).

## 5 Conclusions

The litter decomposition process is a typical example of how relatively small-scale observational studies are used for model evaluation and validation purposes. As decomposition is a process that develops in temporal stages, different processes are

controlling at different time scales. Our results showed that MIMICS+ replicated the relatively long-term CIDET study well, and represented the effects of climate and chemical recalcitrance adequately. However, results from the short-term VCG study were more variable and demonstrated how extra caution should be taken when using small-scale observations to inform soil models, especially in regions that experience regular freezing. Our results showed that locally observed input data did not significantly improve modeled results at the VCG sites. This is likely a combination of the quality of the locally observed

data, experiment design, and model structure and equations. Bridging the gap between empirical field experiments and model development is therefore crucial to fully utilize the information that can be gathered from both scientific fields (Halbritter et al., 2020; Kyker-Snowman et al., 2022). To increase the usefulness of short-term litterbag experiments for the new generation of microbially explicit decomposition models, efforts should be made to include measurements of microbial biomass, stoichiometry, and potential abiotic leaching loss during the experimental period. At the same time, improved representation of snow

cover, and thereby soil freezing processes in land-surface models would improve small-scale model representation of decomposition in cold regions. Together, these measures can advance our understanding of one of the most important processes in the carbon cycle; the decomposition process.



*Code and data availability.* Input data and the MIMICS+ model version used in this study are available at https://doi.org/10.5281/zenodo.
10557583 together with a Jupyter notebook that was used for producing plots and analysis. The VCG litterbag data are available at https:
//osf.io/npfa9/files/osfstorage (7_Ecosystem_data).



## Appendix A: Site information

This appendix provide details about the sites and litter types from the CIDET and VCG experiments.



**Figure A1.** Illustration showing the location of the sites used in this study. a) the regions in Canada (red rectangle) and Norway (blue rectangle), b) location of the nine sites from the CIDET data (see Table A1), c) location of the 12 sites in the Vestland Climate Grid (see Table A3).



**Table A1.** Information about sites from the CIDET study. Data are collected from Tables 1 and 4 in Trofymow and the CIDET Working Group (1998). The remaining mass and standard errors of mean (in parantheses) averaged across all litter types after 3 and 6 years are from Moore et al. (1999) and Trofymow et al. (2002), respectively.

| Short name | Long name | Lat. | Lon. | MAP | MAT | JUN-SEPT | Slope | % mass remaining after: | |
|---|---|---|---|---|---|---|---|---|---|
| | | °N | °E | $mm$ | °$C$ | °$C$ | % | 3 years | 6 years |
| HID | Hidden Lake | 50.6 | -118.8 | 547 | 6.3 | 15.6 | 0.00 | 59.1 (4.5) | 35 (4.1) |
| MAR | Morgan Arb. | 45.4 | -73.9 | 863 | 6.1 | 18.5 | 0.00 | 44.1 (4.0) | 24 (2.3) |
| PMC | Port McNeill | 50.6 | -127.3 | 1783 | 7.9 | 12.7 | 3.50 | 46.5 (5.7) | 35 (4.1) |
| SHL | Shawnigan Lake | 48.6 | -123.7 | 1215 | 9.3 | 15.8 | 0.00 | 43.4 (3.7) | 28 (2.6) |
| SCH | Schefferville | 54.9 | -66.7 | 769 | -4.8 | 9.3 | 0.00 | 70.4 (5.0) | 53 (5.2) |
| NEL | Nelson House1 | 55.9 | -98.6 | 542 | -3.9 | 12.2 | 5.00 | 79.8 (4.3) | 59 (4.2) |
| MON | Montmorency | 47.3 | -71.1 | 1494 | 0.6 | 12.5 | 8.00 | 60.0 (5.7) | 37 (3.3) |
| GIL | Gillam1 | 56.3 | -94.8 | 485 | -5.2 | 11.4 | 1.00 | 80.1 (4.5) | 65 (4.8) |
| CHA | Chaplau | 47.6 | -83.2 | 834 | 1.1 | 14.2 | 0.00 | 52.8 (4.7) | 30 (1.7) |

**Table A2.** Information about litter types from the CIDET study. Data are collected from Table 3 in Trofymow and the CIDET Working Group (1998). The $f_{met}$ values are calculated using the lignin:N data from this table in Eq. B4 in Appendix B.

| Common name | Type | C | N | C:N | Klason lignin | lignin:N | $f_{met}$ |
|---|---|---|---|---|---|---|---|
| (*Scientific name*) | | $mg/g$ | $mg/g$ | - | $mg/g$ | - | - |
| Douglas fir (*Pseudotsuga menziesii*) | needles | 496.1 | 7 | 70.9 | 303 | 43.3 | 0.25 |
| Tamarack (*Larix laricina*) | needles | 487.5 | 5.9 | 82.6 | 240 | 40.6 | 0.25 |
| Jack pine (*Pinus banksiana*) | needles | 497.2 | 12.8 | 38.8 | 328 | 25.6 | 0.39 |
| Black spruce (*Picea mariana*) | needles | 494.7 | 7.3 | 67.8 | 283 | 38.7 | 0.26 |
| Western redcedar (*Thuja plicata*) | needles | 496.7 | 6.4 | 77.6 | 356 | 55.5 | 0.25 |
| American beech (*Fagus grandifolia*) | leaves | 470 | 7.1 | 66.2 | 280 | 39.4 | 0.25 |
| White birch (*Betula papyrifera*) | leaves | 480 | 7.2 | 66.7 | 240 | 33.3 | 0.31 |
| Trembling aspen (*Populus tremuloides*) | leaves | 468.3 | 6.7 | 69.9 | 144 | 21.4 | 0.43 |
| Bracken fern (*Pteridium aquilinum*) | fern | 463.3 | 8.8 | 52.6 | 329 | 37.4 | 0.27 |
| Plains rough fescue (*Festuca halii*) | grass | 437.9 | 7.1 | 61.7 | 112 | 15.7 | 0.48 |
| Western hemlock (*Tsuga heterophylla*) | wood block | 473.3 | 2.4 | 197.2 | 294 | 122.6 | 0.25 |



**Table A3.** Information about the Vestland Climate Grid sites and litter decomposition experiment (Vandvik et al., 2022; Telford et al., 2023). The site names indicate temperature and precipitation levels; alpine, sub-alpine, and boreal along a precipitation gradient from low (1) to high (4)

| Site | Lat | Lon | MAP | JUN-SEPT | LIT C:N | % mass remaining after: | | | |
|------|-----|-----|-----|----------|---------|---------|----------|----------|--------|
|      | °N  | °E  | mm/yr | deg C |         | 1 month | 2 months | 3 months | 1 year |
| Alpine |   |   |   |   |   |   |   |   |   |
| ALP1 | 61.0 | 8.1 | 1226 | 7.0 | 25.2 | 93.4 | 92.6 | 83.9 | 66.5 |
| ALP2 | 60.8 | 7.3 | 1561 | 7.0 | 42.1 | 94.7 | 92.4 | 83.5 | 78.4 |
| ALP3 | 60.8 | 7.2 | 2130 | 6.5 | 49.3 | 84.9 | 89.0 | 78.7 | 74.7 |
| ALP4 | 60.9 | 6.4 | 3402 | 7.0 | 31.8 | 89.9 | 91.5 | 75.4 | 56.1 |
| Sub-alpine |   |   |   |   |   |   |   |   |   |
| SUB1 | 60.8 | 8.7 | 789 | 9.0 | 36.7 | 95.1 | 82.6 | 65.8 | 31.5 |
| SUB2 | 60.9 | 7.2 | 1356 | 9.0 | 34.6 | 96.2 | 77.6 | 77.5 | 47.4 |
| SUB3 | 61.1 | 6.6 | 1848 | 8.7 | 47.9 | 96.6 | 87.5 | 81.2 | 80.1 |
| SUB4 | 60.5 | 6.5 | 3029 | 8.6 | 53.9 | 98.1 | 91.0 | 91.8 | 90.1 |
| Boreal |   |   |   |   |   |   |   |   |   |
| BOR1 | 61.0 | 9.1 | 600 | 10.3 | 37.7 | 93.2 | 77.2 | 53.1 | 35.8 |
| BOR2 | 60.9 | 7.2 | 1161 | 10.5 | 30.4 | 90.4 | 73.3 | 51.5 | 32.3 |
| BOR3 | 60.7 | 6.3 | 2044 | 10.6 | 42.5 | 95.3 | 85.2 | 84.7 | 79.0 |
| BOR4 | 60.7 | 6.0 | 2923 | 10.8 | 41.8 | 94.1 | 81.6 | 80.0 | 62.4 |



## Appendix B: Model details

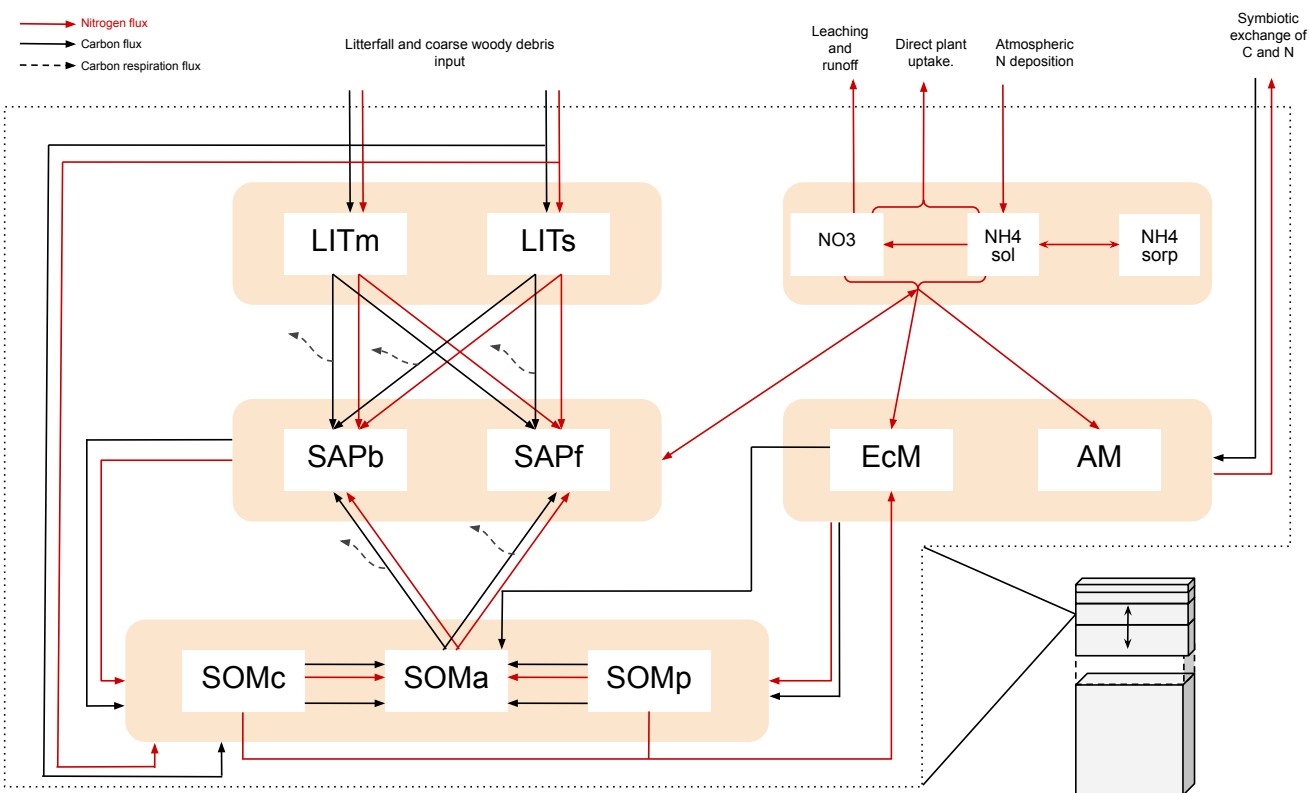

**Figure B1.** Figure adopted from Aas et al. (2023). Schematic showing C and N flows within each layer of MIMICS+. Black arrows indicate carbon fluxes ($gCm^{-3}h^{-1}$) while red arrows indicate nitrogen fluxes ($gNm^{-3}h^{-1}$). The dashed black arrows symbolize C leaving the system as heterotrophic respiration. LITm, LITs: metabolic and structural litter. SAPb, SAPf: saprotrophic bacteria and fungi. EcM, AM: ecto- and arbuscular mycorrhizal fungi. SOMc, SOMa, SOMp: chemically protected, available and physically protected soil organic matter. $NO_3$, $NH4_{sol}$, $NH4_{sorb}$: Inorganic N in the form of $NO_3$, $NH_4$ in solution and $NH_4$ sorbed to particles, respectively.

## B1   reverse Michaelis-Menten kinetic parameters

$$V_{max} = exp(V_{slope} \cdot T + V_{int}) \cdot a_V \cdot V_{mod} \cdot f(\Theta) \tag{B1}$$

$V_{max}$ is the maximum reaction velocity with units $mg(mg)^{-1}h^{-1}$, T represents temperature and $V_{slope}$ and $V_{int}$ are parameters determined in German et al. (2012). The parameters $a_V \cdot V_{mod}$ can be modified depending on type of substrate an microbial group, see details and values in Aas et al. (2023). $f(\Theta)$ is the moisture modifier described below (Eq. B3).

$$K_m = exp(K_{slope} \cdot T + K_{int}) \cdot a_K \cdot K_{mod} \tag{B2}$$



$K_m$ is the half saturation constant with units $mgCcm^{-3}$, and the other parameters served the same purpose as described for $V_{max}$.

## B2    Moisture function

$$f(\Theta) = max\left(0.05, P \cdot \left(\frac{\Theta_{liq}}{\Theta_{sat}}\right)^3 \cdot \left(1 - \frac{\Theta_{liq}}{\Theta_{sat}} - \frac{\Theta_{frozen}}{\Theta_{sat}}\right)^{2.5}\right) \tag{B3}$$

$\Theta_{liq}$ is soil water in liquid form, $\Theta_{ice}$ is soil water in ice form and $\Theta_{sat}$ is saturated soil water content (porosity). $f(\Theta)$ have a

minimum value of 0.05 and a maximum value of 1, the P parameter normalizes the function. The first parenthesis, raised to the power of 3 represent inhibition at low soil moisture, while the second parenthesis represent the oxygen diffusivity limitation at high soil moisture values as a power-law function (2.5) of air-filled-porosity (Sulman et al., 2014, Supplement).

## B3    Calculation of metabolic fraction, $f_{met}$

Calculation and parameters used in CLM, based on Wieder et al. (2015)

$f_{met} = 0.75 \cdot (0.85 - 0.013 \cdot min(40, \text{lignin} : \text{N})) \tag{B4}$



## Appendix C: Additional figures



**Figure C1.** Left axis: Moisture function values in soil layer where litter is added from the four microclimate sources DEF, MOD ,SP and OBS. Values are smoothed by using a 7-day rolling average. Right axis: remaining litter from observations (black triangles), simulations using DEF microclimate (red pentagons), MOD microclimate (blue squares), SP microclimate (green circles) and OBS microclimate (orange diamonds). The solid vertical line mark the burial date of the litterbag while the dotted vertical lines mark collection dates after 1,2,3 and 12 months.





**Figure C2.** Left axis: Ratio of liquid water to saturated water content from the four microclimate sources DEF, MOD ,SP and OBS. Values are smoothed by using a 7-day rolling average. Right axis: remaining litter from observations (black triangles), simulations using DEF microclimate (red pentagons), MOD microclimate (blue squares), SP microclimate (green circles) and OBS microclimate (orange diamonds). The solid vertical line mark the burial date of the litterbag while the dotted vertical lines mark collection dates after 1,2,3 and 12 months.





**Figure C3.** Left axis: Ratio of frozen water to saturated water content from the four microclimate sources sources DEF, MOD ,SP and OBS. Values are smoothed by using a 7-day rolling average. Right axis: remaining litter from observations (black triangles), simulations using DEF microclimate (red pentagons), MOD microclimate (blue squares), SP microclimate (green circles) and OBS microclimate (orange diamonds). The solid vertical line mark the burial date of the litterbag while the dotted vertical lines mark collection dates after 1,2,3 and 12 months.



**Figure C4.** Remaining mass [%] in litterbags at VCG sites [Vandvik et al., 2022] for MIMICS+ simulations without any moisture limitation on decomposition (solid lines) and observations (dots).





## Appendix D: Surface properties for CLM simulations

This appendix contains tables with information about surface parameters used in the different CLM simulations used in this study.

**Table D1.** Surface parameters used in the CLM simulations of the nine CIDET sites.

| SITE | Dominating PFT | pct dom. PFT | Organic Matter | Slope | Porosity | zbedrock | Sand | Clay |
|------|----------------|--------------|----------------|-------|----------|----------|------|------|
| | | % | $kg \cdot m^{-3}$ | degrees | $mm^3 mm^{-3}$ | m | % | % |
| CHA | needleleaf_evergreen_temperate_tree | 71.0 | 46.2 | 0.7 | 0.59 | 11.6 | 59.0 | 10.0 |
| NEL | needleleaf_evergreen_boreal_tree | 77.0 | 116.5 | 0.2 | 0.86 | 21.7 | 31.0 | 40.0 |
| GIL | needleleaf_evergreen_boreal_tree | 41.0 | 122.2 | 0.2 | 0.88 | 48.1 | 36.0 | 23.0 |
| HID | needleleaf_evergreen_boreal_tree | 53.0 | 38.8 | 8.0 | 0.56 | 1.1 | 60.0 | 10.0 |
| MON | needleleaf_evergreen_temperate_tree | 44.0 | 43.0 | 2.3 | 0.58 | 1.2 | 59.0 | 10.0 |
| SCH | c3_arctic_grass | 49.0 | 89.7 | 0.5 | 0.76 | 14.9 | 59.0 | 25.0 |
| SHL | needleleaf_evergreen_temperate_tree | 94.0 | 52.6 | 4.8 | 0.62 | 2.0 | 51.0 | 11.0 |
| PMC | needleleaf_evergreen_temperate_tree | 91.0 | 61.6 | 6.1 | 0.65 | 1.2 | 52.0 | 11.0 |
| MAR | c3_non-arctic_grass | 43.0 | 30.6 | 0.6 | 0.56 | 32.3 | 34.0 | 28.0 |

**Table D2.** Surface parameters used in the CLM simulations of the 12 VCG sites using default parameters (simulation set DEF, see Fig. 1).

| SITE | Dominating PFT | pct dom. PFT | Organic Matter | Slope | Porosity | zbedrock | Sand | Clay |
|------|----------------|--------------|----------------|-------|----------|----------|------|------|
| | | % | $kg \cdot m^{-3}$ | degrees | $mm^3 mm^{-3}$ | m | % | % |
| ALP1 | c3_arctic_grass | 61.00 | 42.60 | 6.2 | 0.58 | 0.80 | 53.00 | 15.00 |
| ALP2 | c3_arctic_grass | 61.00 | 42.60 | 6.2 | 0.58 | 0.80 | 53.00 | 15.00 |
| ALP3 | c3_arctic_grass | 61.00 | 42.60 | 6.2 | 0.58 | 0.80 | 53.00 | 15.00 |
| ALP4 | c3_arctic_grass | 42.00 | 47.60 | 7.6 | 0.60 | 0.80 | 53.00 | 15.00 |
| SUB1 | c3_arctic_grass | 41.00 | 88.50 | 4.6 | 0.76 | 1.00 | 50.00 | 20.00 |
| SUB2 | c3_arctic_grass | 61.00 | 42.60 | 6.2 | 0.58 | 0.80 | 53.00 | 15.00 |
| SUB3 | c3_arctic_grass | 42.00 | 47.60 | 7.6 | 0.60 | 0.80 | 53.00 | 15.00 |
| SUB4 | c3_arctic_grass | 42.00 | 47.60 | 7.6 | 0.60 | 0.80 | 53.00 | 15.00 |
| BOR1 | c3_arctic_grass | 41.00 | 88.50 | 4.6 | 0.76 | 1.00 | 50.00 | 20.00 |
| BOR2 | c3_arctic_grass | 61.00 | 42.60 | 6.2 | 0.58 | 0.80 | 53.00 | 15.00 |
| BOR3 | c3_arctic_grass | 42.00 | 47.60 | 7.6 | 0.60 | 0.80 | 53.00 | 15.00 |
| BOR4 | c3_arctic_grass | 42.00 | 47.60 | 7.6 | 0.60 | 0.80 | 53.00 | 15.00 |





**Table D3.** Surface parameters used in the CLM simulations of the 12 VCG sites using modified parameters (simulation set MOD, see Fig. 1).

| SITE | Dominating PFT | pct dom. PFT | Organic Matter | Slope | Porosity | zbedrock | Sand | Clay |
|------|---------------|--------------|----------------|-------|----------|----------|------|------|
| | | % | $kg \cdot m^{-3}$ | degrees | $mm^3 mm^{-3}$ | $m$ | % | % |
| ALP1 | c3_non-arctic_grass | 100.00 | 46.20 | 26.00 | 0.60 | 0.40 | 53.00 | 15.00 |
| ALP2 | c3_non-arctic_grass | 100.00 | 29.60 | 27.40 | 0.53 | 0.40 | 53.00 | 15.00 |
| ALP3 | c3_non-arctic_grass | 100.00 | 56.50 | 19.20 | 0.63 | 0.40 | 53.00 | 15.00 |
| ALP4 | c3_non-arctic_grass | 100.00 | 21.00 | 24.00 | 0.50 | 0.40 | 53.00 | 15.00 |
| SUB1 | c3_non-arctic_grass | 100.00 | 28.10 | 21.30 | 0.53 | 0.40 | 50.00 | 20.00 |
| SUB2 | c3_non-arctic_grass | 100.00 | 43.80 | 23.50 | 0.59 | 0.40 | 53.00 | 15.00 |
| SUB3 | c3_non-arctic_grass | 100.00 | 36.40 | 22.10 | 0.56 | 0.40 | 53.00 | 15.00 |
| SUB4 | c3_non-arctic_grass | 100.00 | 45.80 | 20.40 | 0.59 | 0.40 | 53.00 | 15.00 |
| BOR1 | c3_non-arctic_grass | 100.00 | 51.80 | 22.50 | 0.62 | 0.40 | 50.00 | 20.00 |
| BOR2 | c3_non-arctic_grass | 100.00 | 72.00 | 29.70 | 0.69 | 0.40 | 53.00 | 15.00 |
| BOR3 | c3_non-arctic_grass | 100.00 | 49.50 | 21.40 | 0.61 | 0.40 | 53.00 | 15.00 |
| BOR4 | c3_non-arctic_grass | 100.00 | 32.10 | 27.60 | 0.54 | 0.40 | 53.00 | 15.00 |



## Appendix E: Calculations for analysis

### E1  Calculation of relative litter mass loss

The relative mass loss of a litter type compared to Western hemlock (the slowest decomposing substrate) shown in Fig. 3 and Fig. E1 are calculated using the following equation:

$$pct_{relative,i,t} = 100 \cdot (\frac{pct_{lost,i,t} - pct_{lost,whw,t}}{pct_{lost,whw,t}}) \tag{E1}$$

where $pct_{lost,i,t}$ and $pct_{lost,whw,t}$ is the percentage of mass lost of litter type $i$ and Western hemlock, respectively, at time $t$. The figures show yearly average values.

### E2  Calculation of normalized RMSE for mass loss at VCG

The Normalized Root Mean Square Errors (NRMSE) referred to in Section 4.3 are calculated using the following equation

$$NRMSE = \frac{1}{O_{max} - O_{min}} \cdot \sqrt{\frac{1}{N} \cdot \sum_{n=1}^{N}(O_n - M_n)^2} \tag{E2}$$

where $O_{max}$ and $O_{min}$ are maximum and minimum observed percentage mass remaining across N = 48 data points (four sampling times at 12 sites), respectively. M is the corresponding modeled percentage of mass remaining.





**Figure E1.** Yearly average relative mass loss (%) compared to mass loss of the slowest decomposing substrate, western hemlock, at the CIDET sites. Details about sites and litter types are found in Appendix A.



*Author contributions.* Conceptualization: ERA,TB, Data curation: ERA , IA, SG, EL, Formal analysis: ERA, Funding acquisition: TA, VV Investigation: ERA, TB, IA Methodology: ERA, TB, Project administration: ERA, Resources: ERA, Software: ERA, HT, EL, Supervision: TB, Validation: ERA, Visualization: ERA , EL, Writing - original draft: ERA, Writing - review & editing: ERA, EL, IA, HT, SG, TB, VV

*Competing interests.* The authors declare that they have no conflict of interest.

*Acknowledgements.* This work has been funded by the University of Oslo and the research council of Norway (RCN) through the research projects: EMERALD (project no. 294948), FUNDER (project no. 315249) and Green Blue (project no. 287490). The simulations were performed on resources provided by Sigma2 - the National Infrastructure for High Performance Computing and Data Storage in Norway, grant nr. NN2806k/NS2806k. HT thanks the support form the Strategic Research Council (SRC) at the Research Council of Finland (#352431).



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
