# Peer review of "Implications of climate and litter quality for simulations of litterbag decomposition at high latitudes"

_EGUsphere, 2024_

## Author Comment (AC1)

Dear Toni Viskari,

Thank you for your thorough feedback on the manuscript. Regarding the CIDET sites you are right, I made the mistake of assuming that the litterbags were buried, when they were, as you correctly pointed out, placed at the surface. I apologize for the oversight, and appreciate you taking the time to go through the material, allowing us to redo the simulations with litter added to the top model layer instead of the third.

Due to your quick submission of comments, we have already had time to set up and run new simulations. At the same time, we also decided to correct the input to the mycorrhizal pools, which where too low due to a bug in the CLM simulations providing the input to MIMICS+ (https://github.com/ESCOMP/CTSM/issues/2120). This means that we have also rerun the simulations for the VCG sites. The consequences of this bug was minor for the results and analysis for the VCG sites, however mass loss decreased slightly at some sites due to increased microbial competition.

Regarding the new CIDET simulations where litterbags were placed in the top soil layer, mass loss was generally higher than when they were put in the third layer, causing a somewhat less fit with the observations, especially at the colder sites. However, the overall conclusions will mainly remain the same in a revised manuscript with the new simulations. See the attached figures, where (a) is identical to Fig. 2 in the original manuscript, and (b) shows results from the new simulations.

In the revised manuscript we will use the results from the original model experiments (where litter was put in the third layer) to discuss consequences of choosing a layer in this kind of discrete-layer model, as well as in the context that litterbags could move vertically during a multi-year experiment

(a) (b)

[Figure]

Below follows answers (in red) to the line-to-line comments (in black) to the manuscript:

Line 25: Bradford et al., 2016; Joly et al., 2023

It is slightly distracting that the same two articles are quoted on two subsequent lines directly above/below each other. Only the first reference is required or split the references between the sentences.

Good point, this line will be rewritten in the revised manuscript.

Line 33: "…litter are buried…"

Kin of related to the larger criticism given to the manuscript before, but there are plenty of litter bag experiments that are not buried.

The sentence will be changed to "Bags containing native or standard litter are either buried or placed on the soil surface, and collected at pre-determined time intervals. From the mass loss one can estimate decomposition rates and investigate relationships related…"

Line 36: "…and methods using standard litter…"

I would recommend first introducing the issue before stating that there have been suggestion on how to overcome it. In the current form it is a bit confusing as my initial concern was that something had been removed before this part.

Agreed, the text will be rewritten following your suggestion in the revised manuscript.

Line 70: "2. Methods"

I'm putting this comment here as I couldn't figure out where exactly to put in the text. Your description of the two experiments is missing central information regarding the properties of the bags. Not just their dimensions, but the size of the holes they have, are they single or two layer, etc.

This isn't necessary information just for accuracy of description, but a huge challenge in comparing information between different litter bag experiments is that the physical differences in the bags do affect the development of the decomposition timeline, especially in multi-annual experiments.

In the revised manuscript we will include information about litter bag dimensions and mesh size for both CIDET and VCG experiments. In addition, the discussion section will be extended to further discuss consequences of the physical properties of the litterbags, like interference by soil fauna etc. There is a difference in mesh size of the bags used at the CIDET sites (0.5 x 0.25 mm) and the VCG sites (0.28 x 0.28 mm) which will be discussed in more detail in the revised manuscript.

Line 80: "…well-established, relatively long-term experiment (hypothesis 1)"

The litter bags in CIDET are collected every year, so you also have access to the decomposition state after the first years. So why are you only doing using the CIDET data to compare the long-term performance?

This is especially pertinent because just looking at the results in Figure 2, I'm not initially convinced that the MIMICS+ model would perform as well in both time frame here as the litter bag experiment sees the largest drop of the mass during the first year before levelling off, something that doesn't appear to happen in most of the model runs. Although would need to see the measurement comparison there to be certain.

We chose to look at mass loss after three and six years because data were immediately available from previous literature for these years (Moore et al. 1999 and Trofymow et al. 2002, respectively). Comparing with one-year mass loss values presented for the sites MAR, PMC and GIL in Moore et al. 2017 (Fig. 2, https://link.springer.com/article/10.1007/s11104-017-3228-7/figures/2) it seems like the model is indeed underestimating initial mass loss also at the CIDET sites.

This tendency is also evident in the VCG simulations, and the poorer model performance at short time scales is discussed in relation to hypothesis 2. In the revised manuscript we will include references to Moore et al. 2017 for discussing short-term performance at the CIDET sites, but don't see the need to include the data points, as short-term performance is mainly discussed in terms of the VCG sites.

Line 97: 2.3 Description of the soil decomposition model, MIMICS+

First, I don't think that comma there between model and MIMICS+ is necessary, is it? You are right, we will remove the comma.

Second, as far as I could tell neither in the model explanation or in appendix B is there any mention how the model was calibrated? Doesn't need to be an exhaustive explanation, just a few lines about the datasets and methods used.

The MIMICS+ model was calibrated against a database of soil profiles from forested sites in Norway (Strand et al. 2016) in Aas et al. 2024. We will include a reference to this in the methods section of the revised manuscript.

Line 194: "This gave initial decomposer…"

Just to confirm, these initial values are for the date when the litter bags were added? Not steady state approximation?

Correct, it will be clarified in the manuscript.

Line 202: "…we assume minor contributions from…"

Do you mean here in the model context? As this is a very assumption to make for multi-year litter bag experiments.

Yes, this is in context of the model setup, where we only consider mass loss in the litter pools. This will be clarified in the revised manuscript. As for the contribution of non-litter to the mass in the litterbag this will be discussed in connection to the mesh-size of the bags (see answer to line 70, "Methods").

Line 306: "Unpublished results from…"

I will admit being puzzled by this reference as I don't understand how you can have unpublished results from a published paper? Also if you are referencing unpublished results here, they need to still accessible somewhere.

The paper with the analysis is in prep., while the data is accessible from OSF (Telford et al. 2023). We would appreciate the editors opinion on this, but for now the sentence have been rewritten to "Results from the VCG sites (Telford et al. 2023, Althuizen et al., in prep.) show…" .

Line 319: "…leaching processes likely also contributed to the observed mass loss."

This part would benefit from a slight reordering. First explain what the model representation of leaching does and does not include before from there shifting to explaining why it might be contributing here.

Agreed, the sentences will be reordered according to your suggestion here.

Additionally, and this is a little bit tricky, I don't think the leaching being a major factor in the discrepancy is a convincing argument based even on the results presented here. For that to be the case, the leaching needs to be insufficient during the first 6-12 months before kicking in correctly? This isn't mean as a discouraging comment, but rather that the rapid decomposition during the first year of almost any decomposition experiment is a well-known challenge for soil carbon decomposition models and it simply appears to be the case also for MIMICS+.

Line 323: "Their experiment suggested..."

But this is in a bit of a contradiction with the VCG results where the initial decomposition drop lasts for longer than the first couple of weeks, isn't it?

The following is an answer to the second part of the comment to line 319 and line 323:

Our argument is that leaching is a contributing factor (among many) to the initial mass loss of litter in the field. As the leaching process is not captured in the model experiments we speculate in how much of the discrepancy between model and observations might be caused by this lack of process representation. An estimate of how much of (and for how long) mass loss that can be attributed to leaching in the field experiment would therefore help to interpret the model result, and determine whether leaching is a major contributor to the discrepancy or not. We will attempt to clarify the argument in the revised manuscript.

We are not quite sure what you mean with this question: "For that to be the case, the leaching needs to be insufficient during the first 6-12 months before kicking in correctly?" please feel free to clarify this.

Line 403: "Litterbag mesh sizes..."

As already noted, the mesh sizes for the different experiment should have been established in the methods section instead of just mentioned all the way here.

See our answer to line 70 "Methods" above. Litterbag specifications will be provided in the method section, and consequences will be discussed in the Discussion section.

*References:*

Aas, E. R., de Wit, H. A., and K. Berntsen, T.: Modeling boreal forest soil dynamics with the microbially explicit soil model MIMICS+ (v1.0), Geosci. Model Dev., 17, 2929–2959, https://doi.org/10.5194/gmd-17-2929-2024 , 2024.

Moore, T.R., Trofymow, J.A., Prescott, C.E. *et al.* Can short-term litter-bag measurements predict long-term decomposition in northern forests?. *Plant Soil* 416, 419–426 (2017). https://doi.org/10.1007/s11104-017-3228-7

Strand, L. T., Callesen, I., Dalsgaard, L., and de Wit, H. A.: Carbon and nitrogen stocks in Norwegian forest soils – the importance of soil formation, climate, and vegetation type for organic matter accumulation, Can. J. Forest Res., 46, 1459–1473, https://doi.org/10.1139/cjfr-2015-0467, 2016.

---

## Author Comment (AC2)

Dear Emma Hauser,

thank you for your comments and feedback to the manuscript. Based on the comments from reviewer 1 we have run new simulations where the litterbags for the CIDET simulations were placed in the top model layer (instead of the third), and at the same time corrected a bug related to the carbon allocated to mycorrhiza in CLM (so simulations for both VCG and CIDET sites have been rerun). See our response to the reviewer 1 comments for details. The main conclusions of the study will remain the same also with the new simulations. The mycorrhizal carbon allocation correction (which gave more C input to the mycorrhizal pools) lead to slightly less mass loss due to increased microbial competition for available N. However, the effect was small, reflecting the limited interaction between mycorrhiza and litter in the model structure. In the future we aim to couple MIMICS+ to above-ground vegetation, which will allow further exploration into the effect of different mycorrhizal types on decomposition mechanisms.

**Line-by-line comments**

**Introduction**

Line 56: Should 'of' be 'and' at the start of the line?

Yes, thank you. This will be corrected.

Line 60: Just something I'm curious about (although I don't know that you need to include this in the manuscript specifically)—I'm familiar with testing of MIMICS in various sites but has MIMICS+ been tested primarily in northern ecosystems to this point or more broadly?

Yes, with MIMICS+ we have focused on northern ecosystems so far, but future plans involve testing in other ecosystems.

**Methods**

Line 111: It's stated previously what rMMK stands for, but the abbreviation isn't included when reverse Michaelis Menten Kinetics is introduced. It would be good to include the abbreviation rMMK in line 100. Good point, we will include the abbrevation in the revised manuscript.

Section 2.3/2.4: Given that CIDET includes many forested sites and the VCG sites are grasslands, did you make any adjustments to the model to reflect potential differences in mycorrhizal types between these ecosystems and could mycorrhizal parameters contribute to any of the differences in the ability of the model to match observations at the Canadian versus Norway sites? From the methods section, it seems like C allocated to mycorrhizae is determined by CLM data--does that generate differences in the portion of AM and EcM for each site?

For this study we have not made any particular adjustments to reflect differences in mycorrhizal associations, as the focus was primarily on decomposition by the saporotrophic pools which directly interacts with litter in the model.

The C allocated to mycorrhiza in MIMICS+ is determined by CLM data. The flux is calculated based on PFT distribution and nutrient availability in the CLM simulations. However, MIMICS+ distribute the C flux between EcM and AM based on a return of

investment function determined by mycorrhizal biomass and nutrient availability in the MIMICS+ simulations, thus independent of the PFT distribution in the CLM simulations. The mycorrhizal pools do not directly interact with the litter pools in MIMICS+ (see Fig. B1), so the effect of mycorrhiza in the model simulations is mainly competition for the same N as the saprotrophs in this study. In the revised manuscript we will clarify the role of mycorrhiza for the model simulations.

Figure 1 caption: In the 2nd line of the figure caption, 'of' after (DEF) should be 'or.' Thanks, it will be corrected!

Line 197: Is the selected model layer a bit deeper than the depths of the empirical litter bags? For VCG the temperature logger is 5 cm deep—is there a reason the authors opted to use the model layer below that and could this decision alter the degree of consistency between the model and the empirical litterbag results? The area of C added also seems larger than the size of empirical litterbags but maybe this is a feature of model resolution?

Given the shallow soils at the VCG sites (< 20 cm), there are uncertainties connected to the exact depths of both the loggers and the litterbags. The model layer was chosen based on the burial depth of the tea bags used in Althuizen et al. 2018, which was ca 8 cm. The selected model layer covers the depths 6-12 cm, which is a bit deeper than the temperature logger which measured at approximately 5 cm. To limit possible inconsistencies between the modeled and field experiment caused by depth differences, we changed the temperature and moisture also in the layer above and below (see line 172 in the preprint), so the modeled "litterbags" are experiencing the measured temperature and moisture in the OBS simulations.

About the area, this is indeed a feature of model resolution. We assume that the model column is 1 m2. Since we are looking at relative mass loss in this study, the area on which we add the litter is of minor importance here.

**Results**

Line 221: Here it's noted that the sites with highest MAT have some of the largest mass loss during the model experiments indicated in Fig. 2, but it's somewhat hard to tell that these are the sites with the warmest MAT given the data displayed in Fig. 2. Opting to show the June-September temperature in the figure makes it seem like the MAR and HID sites are the warmest, so when PMC and SHL were introduced in line 221, I did a double take. Since the results text is focused on MAT rather than summer season temperature, it might be clearer if MAT is used in the figure as well. We chose JUN-SEPT in the original manuscript to make it consistent with the VCG sites (which locations are chosen based on summer temperatures), but we agree that using MAT will make Fig. 2 and the associated discussion clearer. The layout of the figure will be modified in the revised manuscript.

Figure 3: I don't recall the authors making a note of this in their methods, but why did the authors decide to normalize values using hemlock mass loss values rather than plotting simulated cumulative mass loss for each litter type, including hemlock?

The difference between absolute mass loss is relatively small between the litter types (Fig 2.). To highlight the difference in mass loss of litter types relative to each

other, and see the effects of different litter qualities (discussed in section 4.1) we therefore we chose to normalize with the hemlock values.

Figure 6: Should BGC in the top legend be MOD instead? Yes, it should. Thank you!

**Discussion:**

Line 306: Should the Althuizen et al. reference include "in prep" or "in press"?

The sentence have been changed to "Results from the VCG sites (Telford et al. 2023, Althuizen et al., in prep.) show…" but we would appreciate the editors input here, see also response to the comment to line 306 from reviewer 1.

Line 310: Possibly Rocci et al.'s recent publication in SBB would of interest to the authors here?

Rocci, K. S., Cleveland, C. C., Eastman, B. A., Georgiou, K., Grandy, A. S., Hartman, M. D., … & Wieder, W. R. (2024). Aligning theoretical and empirical representations of soil carbon-to-nitrogen stoichiometry with process-based terrestrial biogeochemistry models. *Soil Biology and Biochemistry*, *189*, 109272.

This is a very interesting paper! A reference to this paper will fit well in this part of the manuscript.

Line 404: Due to mesh sizes, could litter at VGC be interacting with mycorrhizal fungi, as well, and therefore would mycorrhizal representation in the model be even more important to replicating field results? Further, do you know if mesh size is the same for CIDET litter bags? I might expect that the underestimation of mass loss by MIMICS+ has more to do with these fauna interactions than microbial stoichiometry (as discussed in the previous section) given numerous studies that show microbial C:N:P to be fairly constrained.

As mentioned above, the mycorrhizal pools in MIMICS+ do not interact directly with the litter pools (although maybe they should..?), so that kind of interaction will not be captured in the model. However, you might be right about mycorrhizal interactions in the field experiments, which could explain some of the discrepancies between model and observed mass loss. In the revised manuscript more focus will be given to the physical properties of the litterbags and possible consequences (e.g. mycorrhzal and fauna interactions). The mesh size is a bit larger for the CIDET sites, something that also will be discussed in more detail in the revised manuscript (see answers to reviewer 1, line 70, line 202, and 403).

Line 408: Soil nutrient availability, microbial communities, and fauna also change with soil depth so differences in depth representation could result in a number of differences affecting decomposition results. It might be worth bringing up a couple of these other actors as well in this paragraph as the authors see fit.

This is a good point. In the revised manuscript we will address this, especially regarding the discrete soil layer approach in the model, and the choice of soil layer to insert the pulse of  litter (litterbag) in the model.

References:

Althuizen, I.H.J., Lee, H., Sarneel, J.M. *et al.* Long-Term Climate Regime Modulates the Impact of Short-Term Climate Variability on Decomposition in Alpine Grassland Soils. *Ecosystems* 21, 1580–1592 (2018). https://doi.org/10.1007/s10021-018-0241-5

---

## Author Response (AR1)

Dear Daniel Goll,

Below follow our responses and manuscript changes to the referee comments. In the revised manuscript, the litterbags at the CIDET sites are added to the top model layer, to more correctly replicate the experiments at the sites, as pointed out and requested by reviewer 1. Figure C5 now show the results from adding the litterbags at the CIDET sites in the 3$^{rd}$ layer. Results and discussion have been updated accordingly, and the choice of model layer is discussed in Sect. 4.4.

As mentioned in our initial response, all simulations (VCG and CIDET) were rerun to correct for the bug in the CLM simulations that provide input to the mycorrhizal pools in MIMICS+. The consequences of this bug was minor for the results and analysis, however mass loss decreased slightly at some sites due to increased microbial competition. This means that all figures have been updated, in addition to updated numerical values in the abstract, results, and discussion.

The updated code and jupyter notebook for analysis is added as a new version to the zenodo repository (line 489).

In addition to changes related to the new simulations and reviewer comments, some minor changes have been done, these are listed at the bottom of this document.

In the following, The line numbers referred to under "Changes in manuscript" below are corresponding to the marked-up manuscript showing the changes.

**Referee 1:**

**Referee comment:**

Line 25: Bradford et al., 2016; Joly et al., 2023

It is slightly distracting that the same two articles are quoted on two subsequent lines directly above/below each other. Only the first reference is required or split the references between the sentences.

**Author's response:**

Good point, this line will be rewritten in the revised manuscript.

**Changes in manuscript:**

The sentences now reads: "The relative importance of climate and litter quality for decomposition rates has recently been debated (Bradford et al., 2016). While macroclimate may generally appear to be the most influential mechanism, Joly et al. (2023) illustrated how macroclimate affects litter quality through plant composition, which further can influence decomposer communities and thus decomposition rates." (line 25-26)

**Referee comment:**

Line 33: "…litter are buried…"

Kin of related to the larger criticism given to the manuscript before, but there are plenty of litter bag experiments that are not buried.

**Author's response:**

The sentence will be changed to "Bags containing native or standard litter are either buried or placed on the soil surface, and collected at pre-determined time intervals. From the mass loss one can estimate decomposition rates and investigate relationships related…"

**Changes in manuscript:**

The text have been changed as described above (line 33-34).

**Referee comment:**

Line 36: "…and methods using standard litter…"

I would recommend first introducing the issue before stating that there have been suggestion on how to overcome it. In the current form it is a bit confusing as my initial concern was that something had been removed before this part.

**Author's response:**

Agreed, the text will be rewritten following your suggestion in the revised manuscript.

**Changes in manuscript:**

The text now reads: "Litterbag experiments often cover limited spatial and temporal scales, and can not be expected to cover all phases of decomposition. Some long-term ( > 10 years) experiments exist (Trofymow and the CIDET Working Group, 1998; Harmon et al., 2009), and methods using standard litter have been proposed to overcome the issue (Keuskamp et al., 2013, Sarneel et al., 2024)". (line 35-39).

**Referee comment:**

Line 70: "2. Methods"

I'm putting this comment here as I couldn't figure out where exactly to put in the text. Your description of the two experiments is missing central information regarding the properties of the bags. Not just their dimensions, but the size of the holes they have, are they single or two layer, etc.

This isn't necessary information just for accuracy of description, but a huge challenge in comparing information between different litter bag experiments is that the physical differences in the bags do affect the development of the decomposition timeline, especially in multi-annual experiments.

**Author's response:**

In the revised manuscript we will include information about litter bag dimensions and mesh size for both CIDET and VCG experiments. In addition, the discussion section will be extended to further discuss consequences of the physical properties of the litterbags, like interference by soil fauna etc. There is a difference in mesh size of the bags used at the CIDET sites (0.5 x 0.25 mm) and the VCG sites (0.28 x 0.28 mm) which will be discussed in more detail in the revised manuscript.

**Changes in manuscript:**

Information about mesh-size and litterbag dimension have been added to lines 75 (CIDET) and 93 (VCG). Implications are discussed in section 4.4 (line 436-445).

**Referee comment:**

Line 80: "…well-established, relatively long-term experiment (hypothesis 1)"

The litter bags in CIDET are collected every year, so you also have access to the decomposition state after the first years. So why are you only doing using the CIDET data to compare the long-term performance?

This is especially pertinent because just looking at the results in Figure 2, I'm not initially convinced that the MIMICS+ model would perform as well in both time frame here as the litter bag experiment sees the largest drop of the mass during the first year before levelling off, something that doesn't appear to happen in most of the model runs. Although would need to see the measurement comparison there to be certain.

**Author's response:**

We chose to look at mass loss after three and six years because data were immediately available from previous literature for these years (Moore et al. 1999 and Trofymow et al. 2002, respectively). Comparing with one-year mass loss values presented for the sites MAR, PMC and GIL in Moore et al. 2017 (Fig. 2, https://link.springer.com/article/10.1007/s11104-017-3228-7/figures/2) it seems like the model is indeed underestimating initial mass loss also at the CIDET sites.

This tendency is also evident in the VCG simulations, and the poorer model performance at short time scales is discussed in relation to hypothesis 2. In the revised manuscript we will include references to Moore et al. 2017 for discussing short-term performance at the CIDET sites, but don't see the need to include the data points, as short-term performance is mainly discussed in terms of the VCG sites.

**Changes in manuscript:**

Based on comments from editor and reviewer, we have included 1-year mass loss for MAR, PMC and GIL from the Moore et al., 2017 article. For a more complete comparison we decided to also include one-year foliar litter mass loss for all sites obtained from data used in Viskari et al., 2022 in Fig. 2 and related discussion. One-year mass loss matched observations quite well for all but the warmest sites, so the model generally perform better for the CIDET sites than the VCG sites after one year. The inclusion of one-year mass loss is mentioned in the abstract (line 8), Sect. 2.1 (line 79-81), 3.1 (line 229), Sect 4.1 (lines 312-316), Sect. 4.2 (lines 326-329, 360).

**Referee comment:**

Line 97: 2.3 Description of the soil decomposition model, MIMICS+

First, I don't think that comma there between model and MIMICS+ is necessary, is it

Second, as far as I could tell neither in the model explanation or in appendix B is there any mention how the model was calibrated? Doesn't need to be an exhaustive explanation, just a few lines about the datasets and methods used.

**Author's response:**

You are right, we will remove the comma.

The MIMICS+ model was calibrated against a database of soil profiles from forested sites in Norway (Strand et al. 2016) in Aas et al. 2024. We will include a reference to this in the methods section of the revised manuscript.

**Changes in manuscript:**

The comma is removed. The following have been added to the first paragraph of section 2.3: "In this study MIMICS+ v1.0 is used, which is calibrated and validated against a database of carbon and nitrogen stocks in soil profiles from forested sites in Norway (Strand et al., 2016)." Model calibration is also discussed in Sect. 4.3 (lines 423-427).

**Referee comment:**

Line 194: "This gave initial decomposer…"

Just to confirm, these initial values are for the date when the litter bags were added? Not steady state approximation?

**Author's response:**

Correct, it will be clarified in the manuscript.

**Changes in manuscript:**

Changed "where" to "at the time the" litterbags were added (line 207).

**Referee comment:**

Line 202: "…we assume minor contributions from…"

Do you mean here in the model context? As this is a very assumption to make for multi-year litter bag experiments.

**Author's response:**

Yes, this is in context of the model setup, where we only consider mass loss in the litter pools. This will be clarified in the revised manuscript. As for the contribution of non-litter to the mass in the litterbag this will be discussed in connection to the mesh-size of the bags (see answer to line 70, "Methods").

**Changes in manuscript:**

Changed the last part of the sentence (line 215) from "and therefore only consider mass in the litter pools." to "and therefore only consider mass in the **model** litter pools." Mesh-size is discussed in lines 436-445.

**Referee comment:**

Line 306: "Unpublished results from…"

I will admit being puzzled by this reference as I don't understand how you can have unpublished results from a published paper? Also if you are referencing unpublished results here, they need to still accessible somewhere.

**Author's response:**

The paper with the analysis is in prep., while the data is accessible from OSF (Telford et al. 2023). We would appreciate the editors opinion on this, but for now the sentence have been rewritten to "Results from the VCG sites (Telford et al. 2023, Althuizen et al., in prep.) show…" .

**Changes in manuscript:**

Based on the editor's suggestion, the sentence now read: "Analysis (Althuizen et al., in prep.) of data from the VCG sites (Telford et al., 2023; Vandvik et al., 2022) show a significant effect…" (lines 332-333)

**Referee comment:**

Line 319: "…leaching processes likely also contributed to the observed mass loss."

This part would benefit from a slight reordering. First explain what the model representation of leaching does and does not include before from there shifting to explaining why it might be contributing here.

**Author's response:**

Agreed, the sentences will be reordered according to your suggestion here.

**Changes in manuscript:**

Deleted the original sentence, and added the following after the explanation of what the model does:

"Leaching processes could have contributed to the observed mass loss, and might explain some of the discrepancies between modeled and observed values." (line 349-350)

**Referee comment:**

Additionally, and this is a little bit tricky, I don't think the leaching being a major factor in the discrepancy is a convincing argument based even on the results presented here. For that to be the case, the leaching needs to be insufficient during the first 6-12 months before kicking in correctly? This isn't mean as a discouraging comment, but rather that the rapid decomposition during the first year of almost any decomposition experiment is a well-known challenge for soil carbon decomposition models and it simply appears to be the case also for MIMICS+.

Line 323: "Their experiment suggested…"

But this is in a bit of a contradiction with the VCG results where the initial decomposition drop lasts for longer than the first couple of weeks, isn't it?

**Author's response:**

The following is an answer to the second part of the comment to line 319 and line 323:

Our argument is that leaching is a contributing factor (among many) to the initial mass loss of litter in the field. As the leaching process is not captured in the model experiments we speculate in how much of the discrepancy between model and observations might be caused by this lack of process representation. An estimate of how much of (and for how long) mass loss that can be attributed to leaching in the field experiment would therefore help to interpret the model result, and determine whether leaching is a major contributor to the discrepancy or not. We will attempt to clarify the argument in the revised manuscript.

We are not quite sure what you mean with this question: "For that to be the case, the leaching needs to be insufficient during the first 6-12 months before kicking in correctly?" please feel free to clarify this.

**Changes in manuscript:**

We added  "However, the extent of mass loss through leaching is hard to estimate, Therefore…" to line 353 to emphasize the argument.

**Referee comment:**

Line 403: "Litterbag mesh sizes…"

As already noted, the mesh sizes for the different experiment should have been established in the methods section instead of just mentioned all the way here.

**Author's response:**

See our answer to line 70 "Methods" above. Litterbag specifications will be provided in the method section, and consequences will be discussed in the Discussion section.

**Changes in manuscript:**

Litterbag specifications are stated in line 75 and 93, and implications are discussed  in section 4.4 (line 436-445).

**Referee 2:**

**Referee comment:**

Line 56: Should 'of' be 'and' at the start of the line?

Figure 1 caption: In the 2nd line of the figure caption, 'of' after (DEF) should be 'or.'

Figure 6: Should BGC in the top legend be MOD instead?

**Changes in manuscript:**

The errors have been corrected.

**Referee comment:**

Line 60: Just something I'm curious about (although I don't know that you need to include this in the manuscript specifically)—I'm familiar with testing of MIMICS in various sites but has MIMICS+ been tested primarily in northern ecosystems to this point or more broadly?

**Author's response:**

Yes, with MIMICS+ we have focused on northern ecosystems so far, but future plans involve testing in other ecosystems.

**Changes in manuscript:**

No changed have been made to the manuscript regarding this comment, as the reviewer indicated that it was not necessary, and we agreed.

**Referee comment:**

Line 111: It's stated previously what rMMK stands for, but the abbreviation isn't included when reverse Michaelis Menten Kinetics is introduced. It would be good to include the abbreviation rMMK in line 100.

**Author's response:**

 Good point, we will include the abbrevation in the revised manuscript.

**Changes in manuscript:**

The abbreviation is introduced in the suggested sentence (line 105) and used in line 118.

**Referee comment:**

Section 2.3/2.4: Given that CIDET includes many forested sites and the VCG sites are grasslands, did you make any adjustments to the model to reflect potential differences in mycorrhizal types between these ecosystems and could mycorrhizal parameters contribute to any of the differences in the ability of the model to match observations at the Canadian versus Norway sites? From the methods section, it seems like C allocated to mycorrhizae is determined by CLM data--does that generate differences in the portion of AM and EcM for each site?

**Author's response:**

For this study we have not made any particular adjustments to reflect differences in mycorrhizal associations, as the focus was primarily on decomposition by the saporotrophic pools which directly interacts with litter in the model.

The C allocated to mycorrhiza in MIMICS+ is determined by CLM data. The flux is calculated based on PFT distribution and nutrient availability in the CLM simulations. However, MIMICS+ distribute the C flux between EcM and AM based on a return of investment function determined by mycorrhizal biomass and nutrient availability in the MIMICS+ simulations, thus independent of the PFT distribution in the CLM simulations. The mycorrhizal pools do not directly interact with the litter pools in MIMICS+ (see Fig. B1), so the effect of mycorrhiza in the model simulations is mainly competition for the same N as the saprotrophs in this study. In the revised manuscript we will clarify the role of mycorrhiza for the model simulations.

**Changes in the manuscript:**

Added a paragraph at the end of section 2.3 (line 145-149) addressing the role of mycorrhiza in the simulations.

**Referee comment:**

Line 197: Is the selected model layer a bit deeper than the depths of the empirical litter bags? For VCG the temperature logger is 5 cm deep—is there a reason the authors opted to use the model layer below that and could this decision alter the degree of consistency between the model and the empirical litterbag results? The area of C added also seems larger than the size of empirical litterbags but maybe this is a feature of model resolution?

**Author's response:**

Given the shallow soils at the VCG sites (< 20 cm), there are uncertainties connected to the exact depths of both the loggers and the litterbags. The model layer was chosen based on the burial depth of the tea bags used in Althuizen et al. 2018, which was ca 8 cm. The selected model layer covers the depths 6-12 cm, which is a bit deeper than the temperature logger which measured at approximately 5 cm. To limit possible inconsistencies between the modeled and field experiment caused by depth differences, we changed the temperature and moisture also in the layer above and below (see line 172 in the preprint), so the modeled "litterbags" are experiencing the measured temperature and moisture in the OBS simulations.

About the area, this is indeed a feature of model resolution. We assume that the model column is 1 m2. Since we are looking at relative mass loss in this study, the area on which we add the litter is of minor importance here.

**Changes in the manuscript:**

Added information about burial depth of the litterbags in section 2.2 (line 93), which corresponds to the chosen model layer.

**Referee comment:**

Line 221: Here it's noted that the sites with highest MAT have some of the largest mass loss during the model experiments indicated in Fig. 2, but it's somewhat hard to tell that these are the sites with the warmest MAT given the data displayed in Fig. 2. Opting to show the June-September temperature in the figure makes it seem like the MAR and HID sites are the warmest, so when PMC and SHL were introduced in line 221, I did a double take. Since the results text is focused on MAT rather than summer season temperature, it might be clearer if MAT is used in the figure as well.

**Author's response:**

We chose JUN-SEPT in the original manuscript to make it consistent with the VCG sites (which locations are chosen based on summer temperatures), but we agree that using MAT will make Fig. 2 and the associated discussion clearer. The layout of the figure will be modified in the revised manuscript.

**Changes in the manuscript:**

The panels in Fig. 2 and C5 (showing the results from when litter was added to the third model layer) is now ordered according to MAT instead of summer temperatures.

**Referee comment:**

Figure 3: I don't recall the authors making a note of this in their methods, but why did the authors decide to normalize values using hemlock mass loss values rather than plotting simulated cumulative mass loss for each litter type, including hemlock?

**Author's response:**

The difference between absolute mass loss is relatively small between the litter types (Fig 2.). To highlight the difference in mass loss of litter types relative to each other, and see the effects of different litter qualities (discussed in section 4.1) we therefore we chose to normalize with the hemlock values.

**Changes in the manuscript:**

Added the following sentence to the beginning of section 3.2: "To investigate the relative difference between the litter types, we compare cumulative mass loss of five litter types to mass loss of the slowest decomposing substrate, western hemlock (Fig. 3, see Fig. C6 for all litter types)."

**Referee comment:**

Line 306: Should the Althuizen et al. reference include "in prep" or "in press"?

**Author's response:**

The sentence have been changed to "Results from the VCG sites (Telford et al. 2023, Althuizen et al., in prep.) show…" but we would appreciate the editors input here, see also response to the comment to line 306 from reviewer 1.

**Changes in the manuscript:**

Based on the editor's suggestion, the sentence now read: "Analysis (Althuizen et al., in prep.) of data from the VCG sites (Telford et al., 2023; Vandvik et al., 2022) show a significant effect…" (lines 332-333)

**Referee comment:**

Line 310: Possibly Rocci et al.'s recent publication in SBB would of interest to the authors here?

Rocci, K. S., Cleveland, C. C., Eastman, B. A., Georgiou, K., Grandy, A. S., Hartman, M. D., ... & Wieder, W. R. (2024). Aligning theoretical and empirical representations of soil carbon-to-nitrogen stoichiometry with process-based terrestrial biogeochemistry models. *Soil Biology and Biochemistry, 189,* 109272.

**Author's response:**

This is a very interesting paper! A reference to this paper will fit well in this part of the manuscript.

**Changes in the manuscript:**

A reference to the suggested publication is added in line 337.

**Referee comment:**

Line 404: Due to mesh sizes, could litter at VGC be interacting with mycorrhizal fungi, as well, and therefore would mycorrhizal representation in the model be even more important to replicating field results? Further, do you know if mesh size is the same for CIDET litter bags? I might expect that the underestimation of mass loss by MIMICS+ has more to do with these fauna interactions than microbial stoichiometry (as discussed in the previous section) given numerous studies that show microbial C:N:P to be fairly constrained.

**Author's response:**

As mentioned above, the mycorrhizal pools in MIMICS+ do not interact directly with the litter pools (although maybe they should..?), so that kind of interaction will not be captured in the model. However, you might be right about mycorrhizal interactions in the field experiments, which could explain some of the discrepancies between model and observed mass loss. In the revised manuscript more focus will be given to the physical properties of the litterbags and possible consequences (e.g. mycorrhzal and fauna interactions). The mesh size is a bit larger for the CIDET sites, something that also will be discussed in more detail in the revised manuscript (see answers to reviewer 1, line 70, line 202, and 403).

**Changes in the manuscript:**

Litterbag specifications are stated in line 75 and 93.  Implications of mesh-size, and possible interactions between litterbag and mycorrhiza are discussed in  lines 436-445.

**Referee comment:**

Line 408: Soil nutrient availability, microbial communities, and fauna also change with soil depth so differences in depth representation could result in a number of differences affecting decomposition results. It might be worth bringing up a couple of these other actors as well in this paragraph as the authors see fit.

**Author's response:**

This is a good point. In the revised manuscript we will address this, especially regarding the discrete soil layer approach in the model, and the choice of soil layer to insert the pulse of  litter (litterbag) in the model.

**Changes in the manuscript:**

Implications of vertical gradients and model layer is discussed in lines 449-458.

**Changes not related to reviewer comments or model reruns:**

The reference to the pre-print Aas et al., 2023 have been changed to the final publication in GMD, Aas et al., 2024.

Line 69: Removed "shift"

The introduction of the VCG abbrevation is introduced in line 65, and removed from line 87.

Line 138-139: Removed "in the litter"

Line 191: Introduced abbrevation "Satellite Phenology"

Line 291: Replaced MAT with "mean annual temperature"

Line 339: Removed sentence deemed unnecessary.

Line 401: "Added generally"

Line: 466-468: Added line about soil moisture optimum.